# Simulating the Climate Response to Atmospheric Oxygen Variability in the Phanerozoic: A Focus on the Holocene, Cretaceous and Permian

David C. Wade[1], Nathan Luke Abraham[1,2], Alexander Farnsworth[3], Paul J. Valdes[3], Fran Bragg[3], and Alexander T. Archibald[1,2]

[1]Centre for Atmospheric Science, Department of Chemistry, Cambridge, UK
[2]National Centre for Atmospheric Science, Department of Chemistry, Cambridge, UK
[3]School of Geographical Sciences, University of Bristol, Bristol, UK

**Correspondence:** dcw32.wade@gmail.com, ata27@cam.ac.uk

**Abstract.** The amount of dioxygen ($O_2$) in the atmosphere may have varied from as little as 5 % to as high as 35 % during the Phanerozoic eon (541 Ma – Present). These changes in the amount of $O_2$ are large enough to have lead to changes in atmospheric mass, which may alter the radiative budget of the atmosphere, leading to this mechanism being invoked to explain discrepancies between climate model simulations and proxy reconstructions of past climates. Here we present the first fully 3D numerical model simulations to investigate the climate impacts of changes in $O_2$ under different climate states using the HadGEM3-AO and HadCM3-BL models. We show that simulations with an increase in $O_2$ content result in increased global mean surface air temperature under conditions of a pre-industrial Holocene climate state, in agreement with idealised 1D and 2D modeling studies. We demonstrate the mechanism behind the warming is complex and involves trade-off between a number of factors. Increasing atmospheric $O_2$ leads to a reduction in incident shortwave radiation at Earth's surface due to Rayleigh scattering, a cooling effect. However, there is a competing warming effect due to an increase in the pressure broadening of greenhouse gas absorption lines and dynamical feedbacks, which alter the meridional heat transport of the ocean, warming polar regions and cooling tropical regions.

Case studies from past climates are investigated using HadCM3-BL which show that in the warmest climate states in the Maastrichtian (72.1–66.0 Ma), increasing oxygen may lead to a temperature decrease, as the equilibrium climate sensitivity is lower. For the Asselian (298.9–295.0 Ma), increasing oxygen content leads to a warmer global mean surface temperature and reduced carbon storage on land, suggesting that high oxygen content may have been a contributing factor in preventing a Snowball Earth during this period of the early Permian. These climate model simulations reconcile the surface temperature response to oxygen content changes across the hierarchy of model complexity and highlight the broad range of Earth system feedbacks that need to be accounted for when considering the climate response to changes in atmospheric oxygen content.

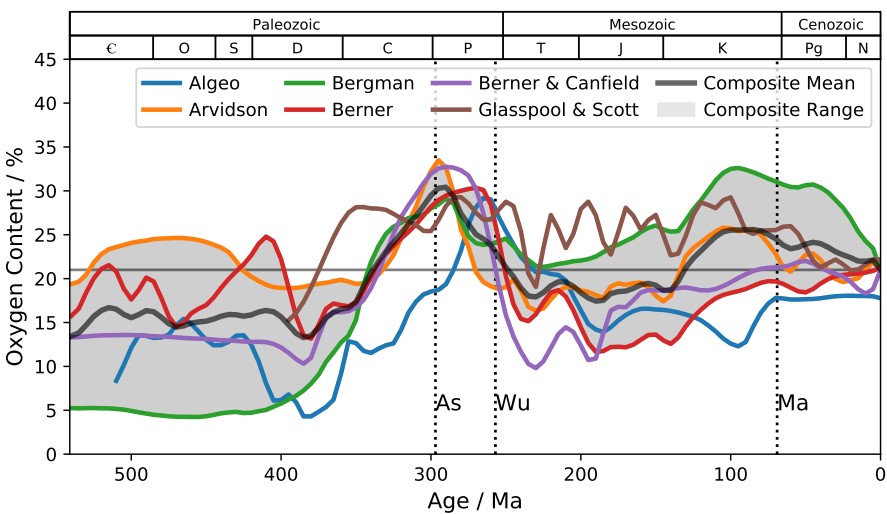

**Figure 1.** Oxygen content reconstructions in the Phanerozoic from Algeo and Ingall (2007), Arvidson et al. (2013), Bergman et al. (2004), Berner (2009), Berner and Canfield (1989) and Glasspool and Scott (2010). The mean (black line) and range (grey shading) of the Arvidson et al. (2013), Bergman et al. (2004) and Berner (2009) is indicated as these reconstructions were most consistent with ice core evidence (Stolper et al., 2016). Present day atmospheric oxygen content is indicated by the horizontal grey solid line. Timings of the palaeo case studies explored in this study are indicated by the vertical dotted lines (As: Asselian, Wu: Wuchiapingian, Ma: Maastrichtian).

## 1    Introduction

The primary driver of climate over the Phanerozoic is atmospheric $CO_2$ (Royer et al., 2004). However, atmospheric oxygen content may also have varied across the Phanerozoic. Atmospheric dioxygen ($O_2$) plays a vital role in the Earth system (Catling et al., 2005), regulating the biosphere through fire ignition (Watson et al., 1978) and metabolism of aerobic biota. Hence

variability in the partial pressure of dioxygen ($pO_2$, a measure of the mass of $O_2$ in the atmosphere, assuming $N_2$ and the volume of the atmosphere have been constant) over time has been invoked as an evolutionary trigger (Berner et al., 2007) of both animals (Falkowski et al., 2005) and plants (He et al., 2012) at many points in the Phanerozoic (Saltzman et al., 2011; Robinson, 1990; Beerling and Berner, 2000; Scott and Glasspool, 2006; Edwards et al., 2017).

While strong biological and geological feedbacks prevent rapid swings in atmospheric oxygen (Catling and Claire, 2005),

reconstructions of past atmospheric oxygen content suggest that there have been substantial excursions from the 21 % oxygen content present in today's atmosphere at times in the Phanerozoic eon. These reconstruction methods can be divided into forward and inversion models. Forward models include nutrient / weathering models (Bergman et al., 2004; Arvidson et al., 2013; Hansen and Wallmann, 2003) and isotope mass balance models (Berner 2009 and Falkowski et al. 2005) while inversion models infer oxygen content from proxies such as charcoal (Glasspool and Scott, 2010), organic carbon to phosphorus ratios

(Algeo and Ingall, 2007) and plant resin $\delta^{13}$C (Tappert et al., 2013). Figure 1 shows the reconstructed oxygen contents for a variety of these methods. There is general agreement in the trends in the reconstructions, in that that oxygen content increased

from 5-25% in the early Paleozoic to 20-35 % in the Permian and subsequently stabilised at levels around 15-30% from the Mid Triassic onward. However, there is uncertainty in the absolute amount of $O_2$ for the different reconstructions (grey shading in Fig. 1). Indeed, there is support for elevated $O_2$ by carbon isotope measurements during the Permian (Beerling et al., 2002). However, disagreement is particularly evident in the Mesozoic, with low values simulated by isotope mass balance approaches.

Mills et al. (2016) have shown that this could be due to an inappropriate choice of $\delta^{13}C$ and that adjusting this value with geological constraints leads to a higher reconstructed oxygen content in better agreement with wildfire records. At the time of writing, there are no direct geochemical proxies for atmospheric $pO_2$ on the Phanerozoic timescale. However, there is isotopic evidence of oceanic oxygenation in steps at approximately 560 (Dahl et al., 2010), 400 (Dahl et al., 2010; Lu et al., 2018) and 200 Mya (Lu et al., 2018). $pO_2$ in the last 800 000 years has been reconstructed using $O_2/N_2$ ratios in ice cores (Stolper et al.,

2016). A roughly 7‰ decline in $pO_2$ is consistent with the ability to change oxygen content by the order of a few percent in ~ 10 Myr. The reconstructions of Bergman et al. (2004), Arvidson et al. (2013) and Berner (2009) are the most plausible based on ice core data (Stolper et al., 2016). Considering these three models alone would still suggest a large uncertainty in oxygen content for most of the Phanerozoic, except for elevated levels in the late Carboniferous / early Permian and reduced levels in the late Devonian.

Phanerozoic means 'visible life' and one of the marked changes to carbon cycling between the Proterozoic and Phanerozoic was caused by the emergence of land plants. The radiation of land plants has led to strong regulation of atmospheric $CO_2$ and $O_2$ which both play important roles in photosynthesis. Land plants likely led to a substantial sequestration of carbon in the terrestrial biosphere and possibly led to the Ordovician glaciation (Lenton et al., 2012). Increases in organic carbon sequestration in the aftermath of the evolution of lignin production may also have contributed to the cooling (Robinson, 1989).

This fundamental change to the Earth system may have constrained $CO_2$ levels to between 10–200 Pa ever since (Franks et al., 2014). Watson et al. (1978) have argued that strong fire feedbacks prevent large fluctuations in oxygen levels, due to runaway burning at high oxygen levels. However subsequent experiments using natural fuels support the possibility of the Earth system to support higher oxygen levels (Wildman et al., 2004). Charcoal appears in the fossil record continuously since the late Devonian (~360 Ma, Algeo and Ingall 2007; Scott and Glasspool 2006). This suggests a floor on oxygen levels in the

region of 12% (Wildman et al., 2004) to 16% (Belcher and McElwain, 2008; Belcher et al., 2010) since then due to limits on ignition.

     Variations in $pO_2$ also have important implications for photosynthesis and therefore the operation of the terrestrial carbon cycle. The primary $CO_2$-fixing enzyme, Rubisco, possesses a dual carboxylase-oxygenase function (Smith, 1976). A photosynthetic carboxylase pathway removes $CO_2$ from the atmosphere while oxygenation leads to photorespiration and $CO_2$

evolution. Therefore, increases in $pO_2$ ought to lead to $O_2$ outcompeting $CO_2$ for active sites on the Rubisco enzyme and leading to a reduction in net primary productivity (less photosynthesis, more respiration). However, photorespiration is likely to be necessary for removal of harmful byproducts in the photosynthetic metabolic pathway (Hagemann et al., 2016) and a recent study suggests that increases in photorespiration may actually promote photosynthesis (Timm et al., 2015). Photosynthesis is itself sensitive to the background $CO_2$ content (Beerling and Berner, 2000). In addition, temperature modifies the relative sol-

ubilities of $CO_2$ and $O_2$ (Jordan and Ogren, 1984). Temperature also affects the specificity of Rubisco for $CO_2$ (Long, 1991).

Therefore, the coevolution of $pO_2$, $pCO_2$ and temperature across the Phanerozoic has the capacity to significantly impact the terrestrial carbon cycle.

This paper focuses on investigating the climate impacts of atmospheric mass variation as the result of altering the concentration of $O_2$. Lower atmospheric mass leads to less Rayleigh scattering so more shortwave radiation reaches the Earth's surface. This enhances atmospheric convection and the hydrological cycle which leads to more tropospheric water vapour, further enhancing warming. However, lower atmospheric mass leads to a reduction in the pressure broadening of greenhouse gas absorption lines which should lead to a weaker greenhouse effect and lead to cooling. Previous modelling studies have investigated which factor dominates with conflicting results. Goldblatt et al. (2009) presented radiative-convective model simulations for the Archean (~3 Ga) which suggested that a nitrogen inventory around three times larger than present would help to keep the early Earth warm at a time when solar input was only around 75% of what it is today, potentially solving the 'Faint Young Sun' paradox (Feulner, 2012). Charnay et al. (2013) investigated this using a GCM coupled to a slab ocean and found that for their idealised early Earth simulations they achieved a strong warming (+7 °C) in response to a doubling in atmospheric mass. Poulsen et al. (2015) simulated the climate impacts of changes in $O_2$ content over a range of 5–35 % using the GENESIS climate model with a slab ocean and a continental configuration consistent with the Cenomanian (mid Cretaceous, 95 Ma) and found the opposite response – lower atmospheric mass at low $pO_2$ was associated with a strong warming. Subsequent 1D calculations cast doubt on this result (Goldblatt, 2016; Payne et al., 2016), however it is plausible that other climate feedbacks such as changes to relative humidity and cloud changes may be important as atmospheric mass changes. These would not be accounted for in 1D radiative-convective equilibrium simulations. Cloud feedbacks in particular are a good candidate for explaining the discrepancy as cloud feedbacks under $CO_2$-driven climate change have strong model dependency (Bony et al., 2015). Another feedback which has not been considered is the possible impact of changes in the mechanical forcing of wind on the ocean circulation. In the absence of this effect, Earth's surface temperature would be 8.7 K cooler (Saenko, 2009) and the equator-to-pole temperature gradient would be steeper. Wind stress ($\tau$) is parameterized in GCMs as $\tau = \rho\, \mathbf{u} \cdot \mathbf{u}$ where $\rho$ is the atmospheric density and $\mathbf{u}$ is the surface wind vector. So as atmospheric density increases, the wind stress on the ocean and therefore ocean heat transport should increase accordingly. Increased meridional heat transport in high density atmospheres is also supported by an idealised 2D modelling study of the early Earth (Chemke et al., 2016). As slab ocean models assume a constant or diffusive ocean heat transport, the Charnay et al. (2013) and Poulsen et al. (2015) simulations cannot account for these effects.

As $pO_2$ variability may alter the radiative budget of the atmosphere, it may also have impacts on the sensitivity of the climate state to $CO_2$ changes. The equilibrium climate sensitivity (ECS) is a metric for the sensitivity of a climate model to an abrupt doubling of atmospheric $CO_2$. Understanding this value is important for predictions of both past and future climatic changes. As the radiative forcing of $CO_2$ is approximately logarithmic with concentration, theoretically the ECS should be constant in time as carbon dioxide changes. However, there is growing evidence that ECS has not been constant over Earth's history (Caballero and Huber, 2013). Changes to the incoming solar radiation (Lunt et al., 2016), palaeogeography (Lunt et al., 2016), $CO_2$ levels themselves (Meraner et al., 2013) and tropical sea-surface temperatures (Caballero and Huber, 2013) may lead to changes in the sensitivity of a particular climate state to changes in $CO_2$.

## 2    Methods & Simulations

### 2.1    Models

The impact of oxygen content variability is investigated with two coupled atmosphere-ocean general circulation models (AOGCMs): HadCM3BL and HadGEM3-AO.

HadGEM3-AO is an AOGCM (Nowack et al., 2014). The atmosphere component is the UK Met Office Unified Model version 7.3 (Davies et al., 2005) in the HadGEM3-A r2.0 climate configuration (Hewitt et al., 2011). It employs a regular Cartesian grid of 3.75° longitude by 2.5° latitude (N48). In the vertical, 60 hybrid height vertical levels are employed – 'hybrid' indicating that the model levels are sigma levels near the surface, changing smoothly to pressure levels near the top of the atmosphere (Simmons and Strüfing, 1983). The model top is 84 km which permits a detailed treatment of stratospheric dynamics. A 20 minute timestep is used. The model employs a non-hydrostatic and fully compressible dynamical core, using a semi-implicit semi-Lagrangian advection scheme on a staggered Arakawa C-grid (Awakawa and Lamb, 1977). Radiation is represented using the Edwards and Slingo (1996) scheme with six short-wave and nine long-wave bands, accounting for the radiative effects of water vapour, carbon dioxide, methane, nitrous oxide and ozone. The MOSES2 land surface scheme is used (Cox et al., 1999) which simulates atmosphere-land exchanges and hydrology. A fixed present-day vegetation distribution of plant functional types is employed. The ocean component of the model is OPA component of the NEMO (Nucleus for European Modelling of the Ocean, Madec 2008) model version 3.0 (Hewitt et al., 2011), run at a 96 minute timestep. In the vertical, 31 model levels are used which increase in thickness steadily between 10 m in the shallowest to 500 m in the deepest layer at 5 km in depth. NEMO employs a tripolar, locally anisotropic grid (ORCA2, Madec 2008) which permits a more detailed treatment of the north polar region and higher resolution in the tropics. This yields an approximate horizontal resolution of 2° in both longitude and latitude, with an increased resolution of up to 0.5° in the tropics. The sea ice component of the model is CICE (Los Alamos Community Ice CodE) at version 4.0 (Hunke and Lipscomb, 2008), run at a 96 minute timestep. This treats sea-ice in a 5-layer model, allowing the simulation of different ice types. The atmosphere and ocean/sea-ice components exchange fields every 24 hours while NEMO and CICE exchange fields every timestep. HadGEM3-AO can be thought of as a close relation to the newest generation HadGEM3 coupled model that will be used to support the next IPCC assessment (Williams et al., 2018) and so represents the state-of-science in numerical climate models.

HadCM3BL (Valdes et al., 2017) is an AOGCM coupled to an interactive vegetation model. The model was originally developed by the United Kingdom Met Office Hadley Centre (Pope et al., 2000) but has since been substantially developed further by the University of Bristol. The atmosphere component of the model employs a regular Cartesian grid of 3.75° longitude by 2.5° latitude. In the vertical, 19 hybrid height vertical levels are employed. A 30 minute timestep is used. The primitive equation set of White and Bromley (1995) is solved to conserve energy and angular momentum, solved on a staggered Awakawa B-grid (Awakawa and Lamb, 1977) in the horizontal. Radiation is represented using the Edwards and Slingo (1996) scheme with six short-wave and eight long-wave bands, accounting for the radiative effects of water vapour, carbon dioxide and ozone, amongst other radiative active species. The ocean component of the model employs the same horizontal grid as the atmosphere component of the model, 3.75° longitude by 2.5° latitude. In the vertical, 20 model levels are used which increase in depth from

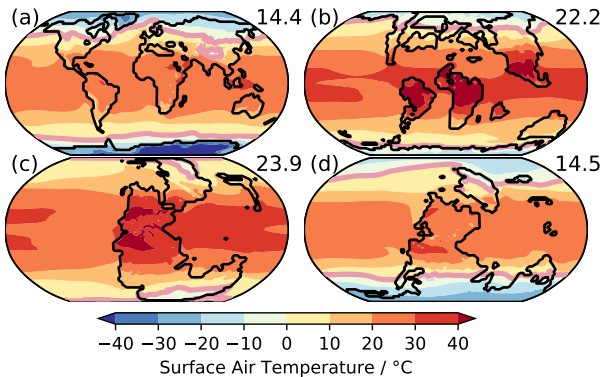

**Figure 2.** Annual average surface air temperature simulated in (a) PI-CM[21], (b) Ma-CM[21], (c) Wu-CM[21] and (d) As-CM[21]. Global mean values are inset top right. The 0 Celsius isoline is indicated in pink. Continental outlines are indicated with a solid black line.

10 m in the shallowest layer to 616 m in the deepest layer. A timestep of 60 minutes is employed and the ocean and atmosphere components exchange required fields once per day. The ocean component is based on the Cox (1984) model, solving the full primitive equation set in three-dimensions. A staggered Awakawa B-grid is employed in both atmosphere and ocean models. Sea-ice is treated as a zero thickness layer on the surface of the ocean grid. Ice is assumed to form at the base at a freezing point
of –1.8 °C but can also form from freezing in ice leads and by falling snow. A simple parameterisation of sea-ice dynamics is also employed (Gordon et al., 2000) and sea-ice formation due to convergence from drift is limited to 4 m thick. Sea-ice albedo is fixed at 0.8 for temperatures below –10 °C, decreasing linearly to 0.5 at 0 °C. The MOSES2.1 land surface model is employed to simulate the fluxes of energy and water between the land surface and the atmosphere (Cox et al., 2000; Essery et al., 2003). TRIFFID (Top-down Representation of Interactive Foliage and Flora Including Dynamics, Cox et al. 1998) predicts the
distribution of vegetation using a plant functional type (PFT) approach. TRIFFID is run in equilibrium mode with averaged fluxes calculated over a 5 year period. TRIFFID calculates vegetation properties for five PFTs: broadleaf trees, needleleaf tree, C3 grass, C4 grass and shrubs. Gridboxes can contain a mixture of PFTs based on a 'fractional coverage co-existence approach' (Valdes et al., 2017). Net primary productivity (NPP) is also calculated, using a photosynthesis-stomatal conductance model (Cox et al., 1998) accounting for a number of factors including atmospheric oxygen content, which affects the photorespiration
compensation point (Clark et al., 2011). The predicted vegetation distribution impacts the atmosphere component by altering surface albedo, evapotranspiration and surface roughness.

## 2.2   Boundary Conditions

Both models simulated the climate response to oxygen variability in a preindustrial Holocene (PIH) climate. HadCM3-BL was additionally run for three time periods across the Phanerozoic: The Maastrichtian (late Cretaceous, 66.0–72.1 Ma), Wuchi-
apingian (late Permian, 254.14–259.1 Ma) and the Asselian (early Permian, 295.0–298.9 Ma). The continental reconstructions employed were developed by and are ©Getech. These reconstructions have been widely employed in a number of previous

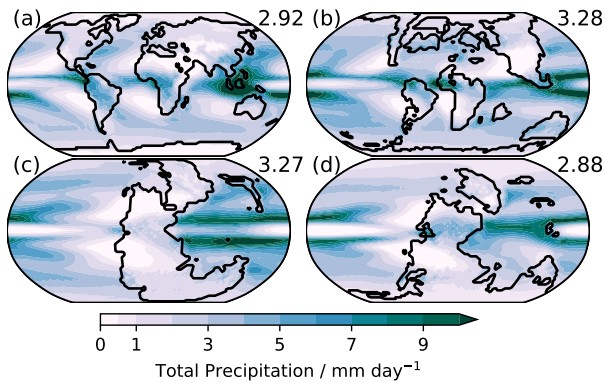

**Figure 3.** Annual average precipitation simulated in (a) PI-CM[21], (b) Ma-CM[21], (c) Wu-CM[21] and (d) As-CM[21]. Global mean values are inset top right. Continental outlines are indicated with a solid black line.

**Table 1.** Summary of experiments. Experiment names AA-BBB include the continental configuration (AA) and model used (BBB). Experiment names NxAA-BBB indicate a multiplier of $CO_2$ with respect to AA-BBB. A star (*) indicates that the $CO_2$ multiplier was applied instantaneously and the transient adjustment to climate was analysed for the purpose of a Gregory et al. (2004) analysis.

| Experiment | Continents | Model | $CO_2$ / Pa | $O_2$ / % |
|------------|------------|-------|-------------|-----------|
| PI-GEM | PIH | HadGEM3-AO | 28 | 10,21,35 |
| 4xPI-GEM | PIH | HadGEM3-AO | 112 | 10,35 |
| PI-CM | PIH | HadCM3-BL | 28 | 10,21,35 |
| Ma-CM | Maastrichtian | HadCM3-BL | 56 | 10,21,35 |
| As-CM | Asselian | HadCM3-BL | 28 | 10,21,35 |
| Wu-CM | Wuchiapingian | HadCM3-BL | 112 | 10,21,35 |
| 2xPI-CM* | PIH | HadCM3-BL | 56 | 10,21,35 |
| 2xMa-CM* | Maastrichtian | HadCM3-BL | 112 | 10,21,35 |
| 2xAs-CM* | Asselian | HadCM3-BL | 56 | 10,21,35 |

studies using the HadCM3-BL climate model (e.g. Lunt et al. 2016). All three are periods of time in which models have suggested that atmospheric oxygen may have deviated significantly from the present level of 21 % (see Fig. 1). Modifications were made to alter the oxygen content of the atmosphere by adjusting the mass mixing ratios of major and minor gases, the surface pressure and other physical characteristics of the atmosphere such as the specific gas constant in an analogous way to Poulsen et al. (2015). Fig. 2 shows the annual average surface temperatures and Fig. 3 shows the annual average precipitation for the (a) PI-CM[21], (b) Ma-CM[21], (c) Wu-CM[21] and (d) As-CM[21] simulations.

A summary of the experiments performed can be found in Table 1. When an experiment with a particular oxygen content is referred to, it will be indicated in superscript, e.g. EXP[21] indicates a 21 % oxygen simulation. 21 % simulations (PI-CM[21],

As-CM[21], Ma-CM[21] and Wu-CM[21]) were integrated for 50 model years as these simulations had already been spun up at that $CO_2$ content. For 10 % and 35 % $pO_2$ the model was spun off the 21 % $pO_2$ simulation and iterated for at least 1000 model years. The 2xPI-CM*, 2xMa-CM* and 2xAs-CM* experiments were spun off from the end of the PI-CM, Ma-CM and As-CM experiments and iterated for 100 years in order to perform a Gregory et al. (2004) analysis. For HadGEM3-AO, model

integrations (PI-GEM[35], PI-GEM[10], 4xPI-GEM[35] and 4xPI-GEM[10]) were performed for 300 model years with a 10-times acceleration of the deep ocean to reduce the time for equilibrium then integrated for a further 500 years to spin up the shallow ocean without acceleration. The last 50 years were used for model analysis.

Pre-Quaternary $pCO_2$ is poorly constrained due to the absence of glacial ice, however there is growing evidence that $CO_2$ is unlikely to have been significantly higher than the order of hundreds of Pa since the radiation of land plants (Breecker

et al., 2010; Franks et al., 2014). For the Maastrichtian, 56 Pa is used in agreement with stomatal proxy-based reconstructions (Steinthorsdottir and Pole, 2016). For the Asselian, 28 Pa is used in agreement with carbonate and fossil plant reconstructions (Montañez et al., 2007). For the Wuchiapingian, 112 Pa is used (Brand et al., 2012).

1D atmospheric chemistry simulations have simulated higher $O_3$ column with increasing $pO_2$ (Kasting et al., 1979; Payne et al., 2016). More detailed 2D model simulations, which capture critical latitudinal gradients in photolysis and zonal mean

transport (Hadjinicolaou and Pyle, 2004; Haigh and Pyle, 1982), support a monotonically increasing ozone column with increasing $pO_2$ (Harfoot et al., 2007). However, simulated ozone column was more sensitive to $N_2O$ levels than $pO_2$ (Harfoot et al., 2007). In addition, while column ozone reduces at low $pO_2$ in Harfoot et al. (2007) there are increases in ozone concentration in the tropical tropopause region where the radiative effect of $O_3$ is stronger (Forster and Shine, 1997). Changes in lightning are important for understanding future changes in tropospheric ozone (Banerjee et al., 2014), however are subject

to considerable uncertainty (Finney et al., 2018). There may be more lightning at high $pO_2$ due to a higher $pO_2/pN_2$ ratio or less due to reduced convection (Goldblatt et al., 2009). Low $pO_2$ may also enhance isoprene emissions (Rasulov et al., 2009), which could enhance tropospheric ozone and alter cloud properties (Kiehl and Shields, 2013). Ozone is also sensitive to changes in $CH_4$ and $N_2O$, the changes to inventories of these chemically-active species on the Phanerozoic timescale (Beerling et al., 2009, 2011) is highly uncertain. Ozone is also sensitive to dynamical changes. Given these large uncertainties in

possible changes to chemical sources, reactivity and transport we neglected including changes in atmospheric ozone concentration in these simulations. However, we recommend that follow up work should focus on this specific question in detail. In HadGEM3-AO the mass of tropospheric and stratospheric ozone is fixed at PIH values simulated by Nowack et al. (2014) using a tropopause height matching scheme. This prevents a rising tropopause leading to stratospheric levels of ozone existing in the troposphere, particularly in the 4xPI-GEM experiments. Not accounting for a rising tropopause has been found to artificially

increase climate sensitivity (Heinemann, 2009) and initial tests not accounting for this led to instability for 4xPI-GEM[10]. In HadCM3-BL, tropospheric ozone is set to 6 ppbv and stratospheric ozone is set to 1.66 ppmv for the 21 % simulations. These values are adjusted to conserve total ozone mass in the alternative $pO_2$ scenarios.

## 2.3 Data

Data for the Cenomanian Poulsen et al. (2015) 21 % $O_2$ and 10 % $O_2$ simulations were obtained from https://www.ncdc.noaa.gov/paleo/study/18776. At the time of writing the 35 % simulation contained missing data so was not used for analysis.

## 2.4 1D Energy Balance Model

A 1D energy balance model (EBM) has been used to deconvolve the contributions from changes in different parts of the climate system. This 1D-EBM approach has been applied to zonal mean quantities for climate simulations of the Eocene by Heinemann et al. (2009) following Budyko (1969) and Sellers (1969):

$$\mathrm{SW}_t^{\downarrow}(\phi)[1-\alpha(\phi)] - \frac{1}{2\pi R^2 \cos(\phi)} \frac{\partial F(\phi)}{\partial \phi} = \epsilon(\phi)\sigma T_{s,\mathrm{ebm}}^4(\phi) \tag{1}$$

where $\mathrm{SW}_t^{\downarrow}$ is the incident shortwave radiation at the top-of-the atmosphere, $\phi$ is the latitude, $\alpha$ is the surface albedo, $R$ is the
radius of Earth, $\epsilon$ is the effective surface emissivity and $T_{s,ebm}$ is the EBM surface temperature (Heinemann et al., 2009). $\frac{\partial F(\phi)}{\partial \phi}$ is the divergence of total meridional heat transport and is given by

$$\frac{\partial F(\phi)}{\partial \phi} = -2\pi R^2 \cos(\phi)(\mathrm{SW}_t^{\mathrm{net}}(\phi) + \mathrm{LW}_t^{\mathrm{net}}(\phi)) \tag{2}$$

where $\mathrm{SW}_t^{\mathrm{net}}$ and $\mathrm{LW}_t^{\mathrm{net}}$ are the net top-of-atmosphere shortwave and longwave radiative fluxes respectively (positive downward, Heinemann et al. 2009). Solving for $T_{s,ebm}$, the EBM surface temperature for each latitude can be calculated using zonal
and annual mean radiative fluxes from the GCM. Where clear-sky radiative fluxes are also available, cloud radiative effects can be deconvolved from clear-sky radiative effects. The clear-sky albedo $\alpha_c$ and clear-sky effective surface emissivity $\epsilon_c$ can be calculated by:

$$\alpha_c = \frac{\mathrm{SW}_{t,c}^{\uparrow}}{\mathrm{SW}_t^{\downarrow}}, \epsilon_c = \frac{\mathrm{LW}_{t,c}^{\uparrow}}{\mathrm{LW}_s^{\uparrow}} \tag{3}$$

where $SW_{t,c}^{\uparrow}$ is the upward top-of-atmosphere clear-sky shortwave radiative flux and $LW_{t,c}^{\uparrow}$ is the upward top-of-atmosphere
clear-sky longwave radiative flux. When considering the temperature change between two experiments, the contributions from different components can be quantified by calculating $T_{s,ebm}$ with different combinations of components from each experiment (Heinemann et al., 2009).

## 2.5 Climate Sensitivity

To estimate the climate sensitivity to $CO_2$ changes, the linear regression methodology of Gregory et al. (2004) is employed.
This assumes a linear relationship between the changes in global, annual mean radiative imbalance at the top-of-atmosphere $(N, \mathrm{Wm}^{-2})$ and surface temperature anomalies $(T_s, {}^{\circ}\mathrm{C})$

$$N = F + \xi \Delta T_s \tag{4}$$

where $\xi$ is the effective climate feedback parameter ($\mathrm{Wm^{-2}\,^{\circ}C^{-1}}$) and $F$ is the effective forcing ($\mathrm{Wm^{-2}}$) accounting for fast climate adjustments and effective radiative forcing. The effective ECS is then $\Delta T_s$ when $N = 0$. While there are weaknesses of this approach, particularly due to non-linearities in $\xi$ as $\Delta T_s$ changes (Armour et al., 2013; Li et al., 2013), the climate response when simulations are continued to equilibrium show an accuracy to within 10 % (Li and Sharma, 2013). Furthermore, the contributions to $\xi$ and $F$ from longwave (LW) and shortwave (SW), clear-sky (CS) and cloudy-sky (CRE) components can be decomposed by a linear decomposition as

$$F = F_{\mathrm{CS,SW}} + F_{\mathrm{CS,LW}} + F_{\mathrm{CRE,SW}} + F_{\mathrm{CRE,LW}} \tag{5}$$

for the effective forcing and

$$\xi = \xi_{\mathrm{CS,SW}} + \xi_{\mathrm{CS,LW}} + \xi_{\mathrm{CRE,SW}} + \xi_{\mathrm{CRE,LW}} \tag{6}$$

for the effective climate feedback parameter.

## 3 Results

Where results are presented from a single simulation, the oxygen content for that run is superscript, i.e. $\mathrm{EXP^{21}}$ indicates a 21 % oxygen simulation. Where results are presented as an anomaly between simulations with different oxygen contents, $\mathrm{EXP^{21}_0}$ indicates that the quantity presented is $\mathrm{EXP^{21}}$ minus $\mathrm{EXP^0}$. A summary of results is shown in Table 2.

### 3.1 Surface Climate

Figure 4 (left) shows the annual-mean surface air temperature differences between the 35 % and 10 % runs. For the preindustrial Holocene, PI-GEM$^{35}_{10}$ (Figure 4a) shows a global mean surface temperature response of $+1.50\,^{\circ}\mathrm{C}$ while PI-CM$^{35}_{10}$ (Figure 4b) shows a similar global mean surface temperature response of $+1.22\,^{\circ}\mathrm{C}$. It is worth noting that HadCM3-BL and HadGEM3-AO are not completely distinct climate models, for instance sharing the Edwards and Slingo (1996) radiation scheme, so this is unlikely to capture the full variability in possible climate model responses. That the results are reasonable agreement with the 1-D results of Payne et al. (2016), who simulated a temperature response between $+1.05$ and $+2.21\,^{\circ}\mathrm{C}$ depending on assumptions about atmospheric ozone, gives some confidence in the HadCM3-BL and HadGEM3-AO results. Similarly, the As-CM$^{35}_{10}$ (Figure 4c) case exhibits a global mean surface temperature response of $+1.29\,^{\circ}\mathrm{C}$. For the warmest climates, a response of $-0.82^{\circ}\mathrm{C}$ is simulated for Wu-CM$^{35}_{10}$ (Figure 4e) and $+0.17\,^{\circ}\mathrm{C}$ for 4xPI-GEM$^{35}_{10}$ (Figure 4f). In the Ma-CM$^{35}_{10}$ case (Figure 4d), a global mean surface temperature response of $+0.70\,^{\circ}\mathrm{C}$ is simulated. This suggests that the climate response to $p\mathrm{O2}$ variability depends on the background climate state.

There is a strong seasonal dependence in the surface air temperature response. Considering the changes in coolest average monthly temperature in each gridbox (Figure 4 middle column), the change in cool month dominates the warming response, particularly at high latitudes. By contrast, the warm month mean is smaller/less negative in all cases (Figure 4 right). A cooling of continental land masses is evident in the tropics and particularly in the Wu-CM (Figure 4e) and 4xPI-GEM (Figure 4f) cases.

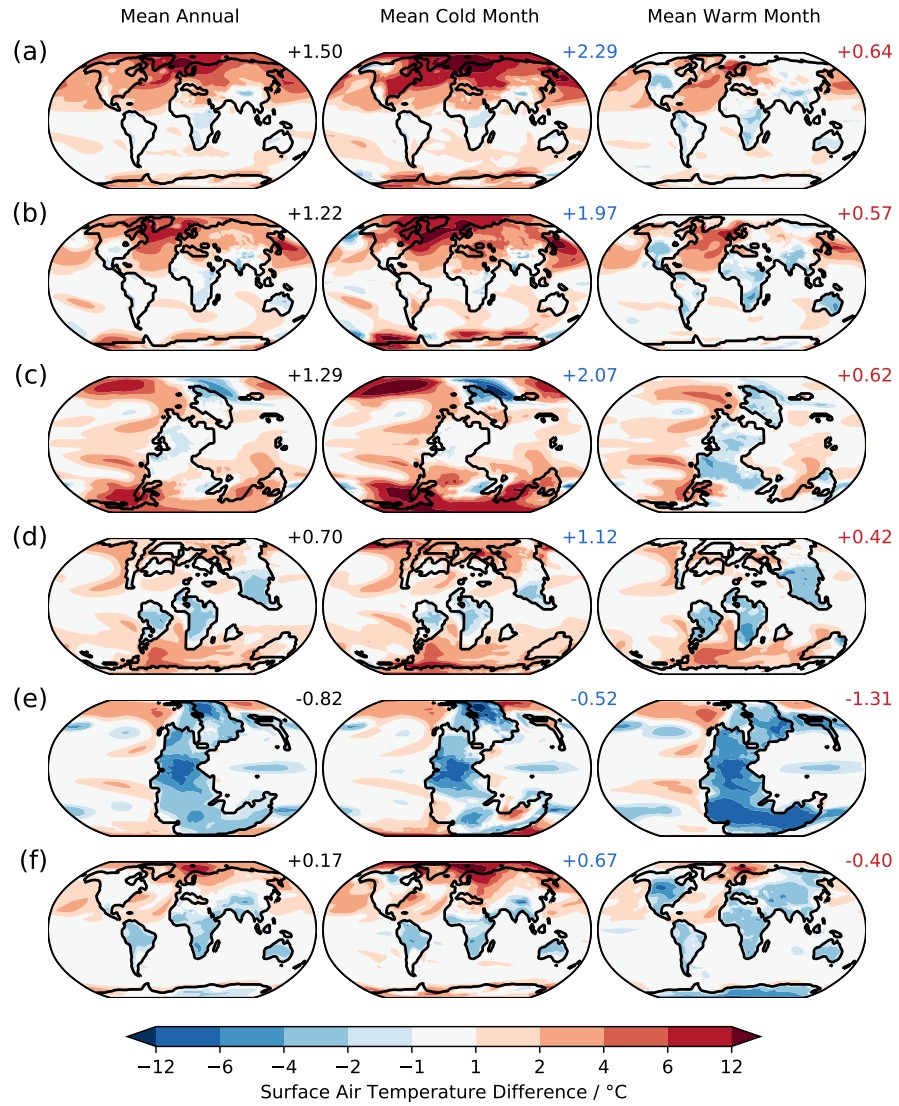

**Figure 4.** Surface air temperature change for (a) PI-GEM$_{10}^{35}$, (b) PI-CM$_{10}^{35}$, (c) As-CM$_{10}^{35}$, (d) Ma-CM$_{10}^{35}$, (e) Wu-CM$_{10}^{35}$ and (f) 4xPI-GEM$_{10}^{35}$ in the annual mean (left), cold month mean (change in the mean gridbox temperature of the coldest month in the monthly mean climatology, middle) and warm month mean (change in the mean gridbox temperature of the warmest month in the monthly mean climatology, right). The change in global mean values (°C) are offset to the top-right of each plot. Note the strong high latitude warming in the cold month mean and tropical cooling in the warm month mean.

**Table 2.** Summary of results for $EXP^{21}$ then $EXP^{35}_{10}$. Where applicable, results calculated for the Poulsen et al. (2015) Cenomanian 21 %–10 % oxygen simulation are also presented. Abbreviations: $T_{\text{eq-pole}}$ (Equator-to-pole surface air temperature gradient), $T_{\text{eq-pole,cold month}}$ (Equator-to-pole surface air temperature gradient for cold month), EBM (quantities obtained using a Budyko-Sellers 1D energy balance model following Heinemann 2009), $T_{\text{s,ebm}}$ (EBM Surface temperature), $T_{\text{s,csky,ebm}}$ (EBM Surface temperature change accounting for changes in clear sky radiative fluxes), $T_{\text{s,cre,ebm}}$ (EBM Surface temperature change accounting for changes in cloudy sky radiative fluxes), $T_{\text{s,mht,ebm}}$ (EBM Surface temperature change accounting for changes in meridional heat flux divergence)

| Quantity | PI-GEM | 4xPI-GEM | PI-CM | As-CM | Ma-CM | Wu-CM | Poulsen |
|---|---|---|---|---|---|---|---|
| $pCO_2$ / Pa | 28 | 112 | 28 | 28 | 56 | 112 | 56 |
| | | | $EXP^{21}$ | | | | $EXP^{21}$ |
| $T_s$ / $^\circ$C | 14.3 | — | 14.4 | 14.5 | 22.2 | 23.9 | 20.5 |
| Precip. / mm day$^{-1}$ | 3.11 | — | 2.92 | 2.88 | 3.28 | 3.27 | 3.49 |
| | | | $EXP^{35}_{10}$ | | | | $EXP^{21}_{10}$ |
| GCM $T_s$ /$^\circ$C | +1.35 | +0.05 | +1.14 | +1.19 | +0.57 | −1.01 | −2.06 |
| $T_{\text{eq-pole}}$ / $^\circ$C | −4.15 | −2.18 | −3.11 | −2.38 | −2.28 | −1.92 | +0.93 |
| $T_{\text{eq-pole,cold month}}$ / $^\circ$C | −6.53 | −4.17 | −5.07 | −5.61 | −3.91 | −2.50 | +1.89 |
| Planetary Albedo | +0.008 | +0.015 | +0.001 | +0.002 | −0.003 | +0.006 | +0.009 |
| Surface Emissivity | −0.019 | −0.011 | −0.011 | −0.013 | −0.003 | +0.004 | +0.008 |
| $T_{\text{s,ebm}}$ / $^\circ$C | +1.30 | +0.10 | +1.10 | +1.08 | +0.53 | −1.01 | −2.05 |
| $T_{\text{s,csky,ebm}}$ / $^\circ$C | +1.51 | +0.05 | +1.45 | +1.35 | +0.90 | −0.30 | −0.60 |
| $T_{\text{s,cre,ebm}}$ / $^\circ$C | −0.43 | −0.13 | −0.57 | −0.49 | −0.36 | −0.58 | −1.25 |
| $T_{\text{s,mht,ebm}}$ / $^\circ$C | +0.22 | +0.18 | +0.22 | +0.22 | −0.01 | −0.13 | −0.20 |

These could be in part due to free-air lapse rate changes which should be stronger at high $pO_2$ as for a given topographic height, the change in pressure is higher for high $pO_2$ which should lead to a larger temperature reduction with height. The changes to the seasonal cycle are consistent with the radiative changes associated with changing oxygen content. The reduction in incident surface shortwave radiation should have its strongest effect on extratropical temperatures in the summer, therefore the Rayleigh

5   scattering component will most strongly affect the warm month temperature. Warming from pressure broadening of greenhouse gas absorption lines as atmospheric mass increases will be most evident in extratropical winter, as with anthropogenic climate change, due to sea-ice and surface heat flux changes (Dwyer et al., 2012). The reduction in the amplitude of the seasonal cycle in temperature simulated by both HadGEM3-AO and HadCM3-BL is therefore supported by a consideration of the changes to atmospheric radiation.

10   The zonal and annual mean surface air temperature changes are shown in Fig. 5. The Northern Hemisphere equator-to-pole temperature gradient is reduced by 6.6 $^\circ$C in the PI-GEM$^{35}_{10}$ case (blue line) and 4.0 $^\circ$C in the PI-CM$^{35}_{10}$ case (not shown). The zonal structure of the surface temperature change is similar in the palaeoclimate case studies. In the Maastrichtian, the equator-to-pole temperature gradient is reduced by 2.0 $^\circ$C (Ma-CM$^{35}_{10}$) and in the Asselian the equator-to-pole temperature gradient is

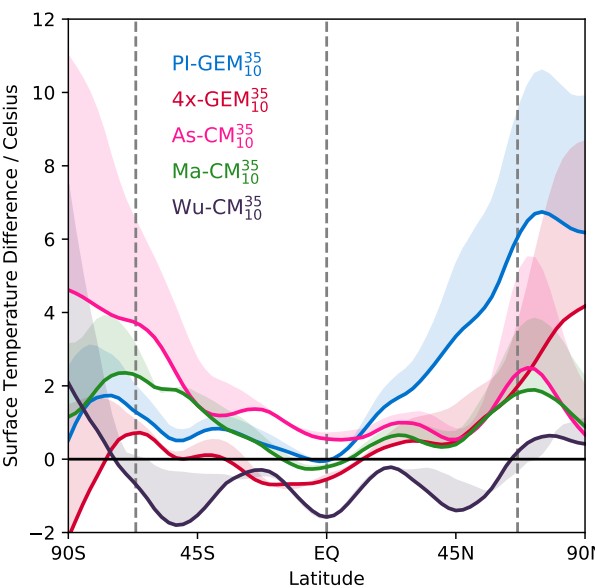

**Figure 5.** Zonally and annually averaged surface air temperature difference (solid lines) from $10\%$ to $35\%$ oxygen content for PI-GEM (blue), 4xPI-GEM (red), As-CM (pink), Ma-CM (green) and Wu-CM (purple). The difference from the annual-mean to cold month-mean for each run is indicated by the shading. Values are smoothed by a Savitzky-Golay filter (Savitzky and Golay, 1964).

reduced by $2.3\,°C$ (As-CM$^{35}_{10}$). The equator-to-pole temperature gradient reduces even in the Wu$^{35}_{10}$ case despite the reduction in global mean surface temperatures.

The hydrological cycle is also affected by changing oxygen content. Increases in Rayleigh scattering at high $p\mathrm{O}_2$ ought to reduce incident shortwave at Earth's surface (Poulsen et al., 2015) and inhibit convection (Goldblatt et al., 2009) which should
lead to reductions in precipitation. This is analogous to stratospheric sulfate or solar radiation management geoengineering where precipitation is reduced in geoengineering experiments with respect to an unperturbed climate with the same global mean surface temperature (Irvine et al., 2016). Poulsen et al. (2015) simulated large reductions in precipitation as $p\mathrm{O}_2$ increased in the GENESIS climate model, however much of this could be explained by the surface temperature changes. Annually averaged precipitation change between the $10\%$ and $35\%$ oxygen content runs are show in Fig. 6. In all cases, increasing oxygen content
leads to a decline in global mean total precipitation, despite the increase in surface temperatures, however with strong regional differences. For PI-GEM (Fig. 6a), PI-CM (Fig. 6b) and 4xPI-GEM (Fig. 6f) there is a clear northward shift in the tropical rain belts. A northward shift in the ITCZ would be consistent with stronger warming in the Northern Hemisphere due to Bjerknes compensation (Bjerknes, 1964; Broccoli et al., 2006). Heat transport is more hemispherically symmetric in the Maastrichtian, Asselian and Wuchiapingian cases so latitudinal ITCZ shifts are not evident. While global precipitation is reduced in Wu-
CM$^{35}$, the increase in ocean-land temperature contrast leads to a significant increase in tropical land precipitation. Despite the increases in global-mean surface temperatures simulated for most cases, precipitation is still reduced in all simulations.

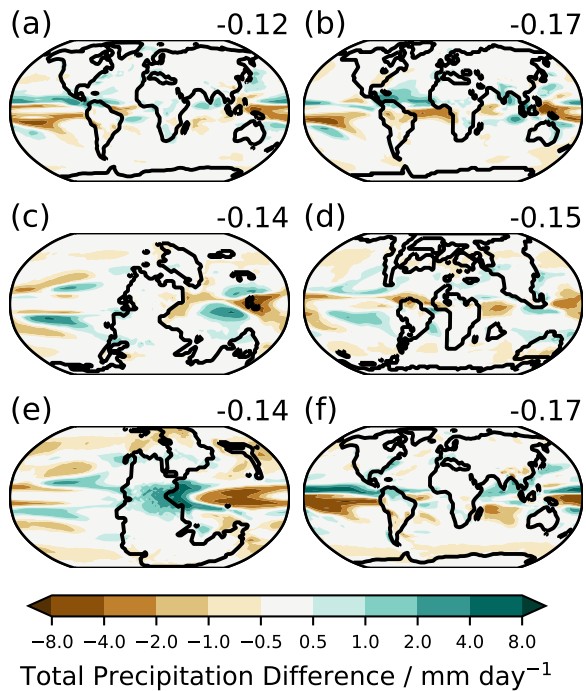

**Figure 6.** Annually averaged total precipitation change from 10 % to 35 % oxygen content for (a) PI-GEM$_{10}^{35}$, (b) PI-CM$_{10}^{35}$, (c) As-CM$_{10}^{35}$, (d) Ma-CM$_{10}^{35}$, (e) Wu-CM$_{10}^{35}$ and (f) 4xPI-GEM$_{10}^{35}$. Global mean values (mm day$^{-1}$) are offset to the top-right of each plot.

Comparing the surface temperature and precipitation response between HadCM3-BL and HadGEM3-AO suggests that the model responses are broadly consistent. A gridbox-by-gridbox comparison of annual mean surface air temperature and precipitation anomalies for PI-GEM$_{10}^{35}$ vs PI-CM$_{10}^{35}$ is presented in Fig. S1. The largest discrepancy in surface air temperature response between the two models occurs for the largest temperature changes simulated by HadGEM, which are strongest in Northern Hemisphere polar regions. This could be linked to differences in the representation of polar climate processes and amplification by polar ice feedbacks between the two models. There is broad consistency in cold and warm-month means (Figure 4a and b) with stronger warming in the cold month mean and terrestrial cooling in the warm month mean.

### 3.2 Energy Balance Decomposition

The drivers of the changes in surface temperature can be understood by decomposing the terms which contribute to surface temperature change in a 1D-energy balance model following Heinemann et al. (2009). For PI-CM$_{10}^{35}$, these results are shown in Fig. 7. These show that the 1D-EBM can reasonably capture the temperature response in the HadCM3-BL simulations, with slight errors (where the black and grey lines are not overlapping) evident in the polar regions. This could be due to averaging over the polar rows in the HadCM3-BL model. There are positive contributions to the surface temperature change in the clear sky emissivity and albedo at the poles. This is consistent with the increase in pressure broadening of absorption

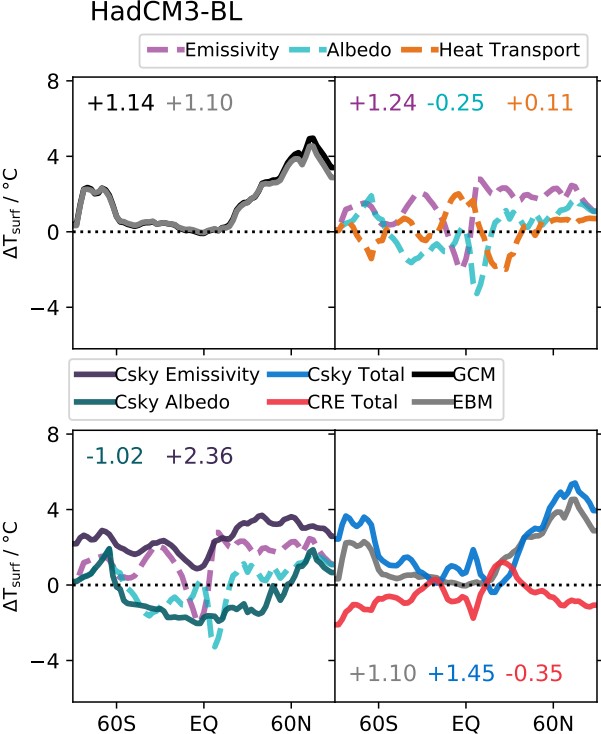

**Figure 7.** 1D-EBM decomposition for PI-CM$^{35}_{10}$. Top left: EBM results (grey) vs GCM results (black). Top right: Decomposition of EBM into the emissivity (purple), albedo (green) and heat transport (orange) components of the temperature change. Bottom left: Clear-sky emissivity (dark purple) and clear-sky albedo (dark green) components of the EBM. The all-sky components are included for comparison. Bottom right: Decomposition of EBM into the total clear-sky (blue), cloudy-sky (red) and all-sky (grey) components.

lines and the simulated reduction in sea-ice extent. By contrast, extrapolar contributions to clear sky albedo provide a negative contribution to the temperature change which is consistent with an increase in Rayleigh scattering which would be expected to be strongest in the tropics where the maximum in incoming solar radiation is located. Combined, the clear sky component of the temperature change is +1.45°C and the cloudy sky component is –0.35°C. This suggests that HadCM3-BL supports a

5  cloud feedback which acts to cool the climate at high $pO_2$ and partially offset the clear-sky temperature changes.

The same analysis was performed for the HadGEM3-AO PIH simulations. Fig. 8 shows that a somewhat weaker cloud feedback is simulated by HadGEM3-AO (–0.21°C). Clear-sky contributions are slightly stronger (+1.51°C). The largest differences between the simulations appear in the all-sky albedo and emissivity changes, where there appear to be competing factors which lead to a similar climate response possibly related to partitioning between the longwave and shortwave contri-

10  butions to the cloud response. This is perhaps unsurprising, as cloud feedbacks to $CO_2$ changes represent a large uncertainty in future climate change projections and given the relatively small global-mean temperature changes a relatively small change in cloud radiative effects has the power to considerably mediate the climate response. However the qualitative agreement in

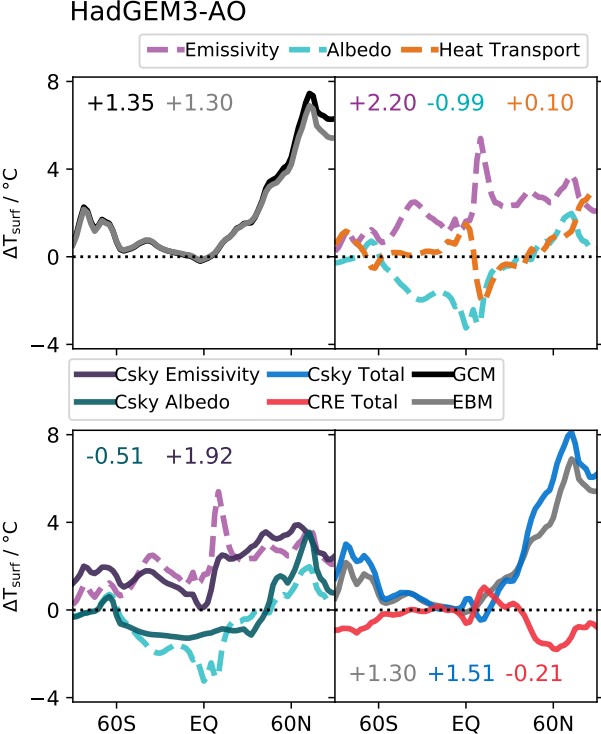

**Figure 8.** 1D-EBM decomposition for PI-GEM$_{10}^{35}$. Top left: EBM results (grey) vs GCM results (black). Top right: Decomposition of EBM into the emissivity (purple), albedo (green) and heat transport (orange) components of the temperature change. Bottom left: Clear-sky emissivity (dark purple) and clear-sky albedo (dark green) components of the EBM. The all-sky components are included for comparison. Bottom right: Decomposition of EBM into the total clear-sky (blue), cloudy-sky (red) and all-sky (grey) components.

latitudinal structure of the clear sky albedo and emissivity changes between these structurally different models gives some confidence that the relevant climate feedbacks are well captured in these simulations.

Analysis of the palaeo-case studies (As-CM, Fig. S2; Ma-CM, Fig. S3; Wu-CM, Fig. S4) shows a similar pattern. In all simulations, irrespective of surface temperature response, the clear sky emissivity is a positive contribution to global mean 5 surface temperature change while clear sky albedo is a more negative contribution. The emissivity contribution becomes less positive as $pCO_2$ increases from As-CM to Ma-CM to Wu-CM. By contrast the albedo contribution becomes more negative as $pCO_2$ increases. This is consistent with the reduction in planetary albedo as sea-ice extent is reduced on the ocean and dark vegetated surfaces increase on the land.

### 3.3 Climate Sensitivity

10 Here we investigate the impact of oxygen variability on climate sensitivity. The HadGEM3-AO and HadCM3-BL results suggest that increasing $CO_2$ content leads to a reduction in the surface temperature change on increasing $pO_2$ (compare 4xPI-

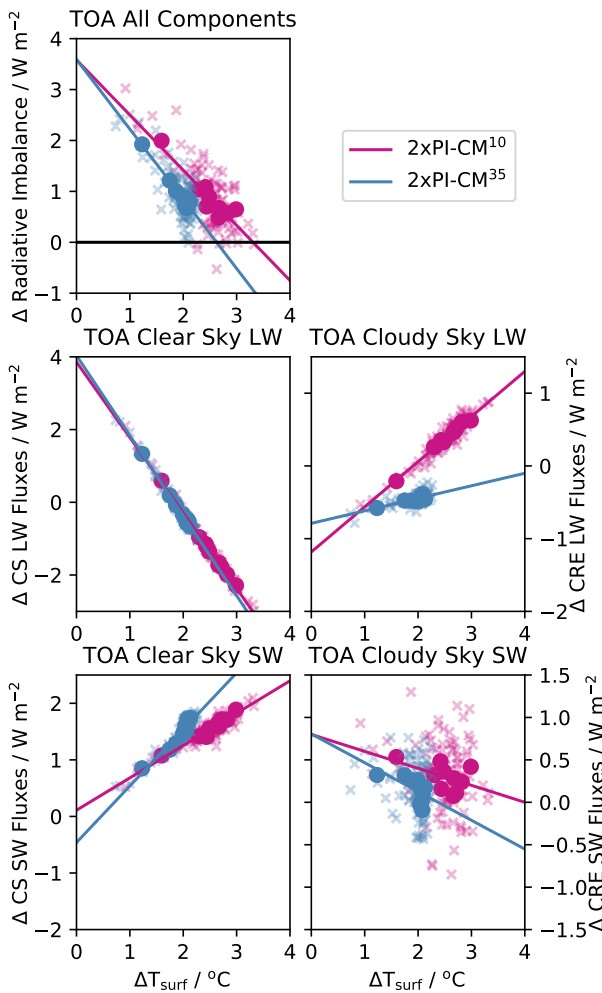

**Figure 9.** Gregory analysis of HadCM3-BL: Regression of top-of-atmosphere radiative imbalance against surface air temperature change (solid lines) for the first 100 years of 2xPI-CM[10] (pink) and 2xPI-CM[35] (blue) cases. Annual averages are indicated by crosses and decadal averages are indicated by filled circles. The regression was performed on the decadal averages.

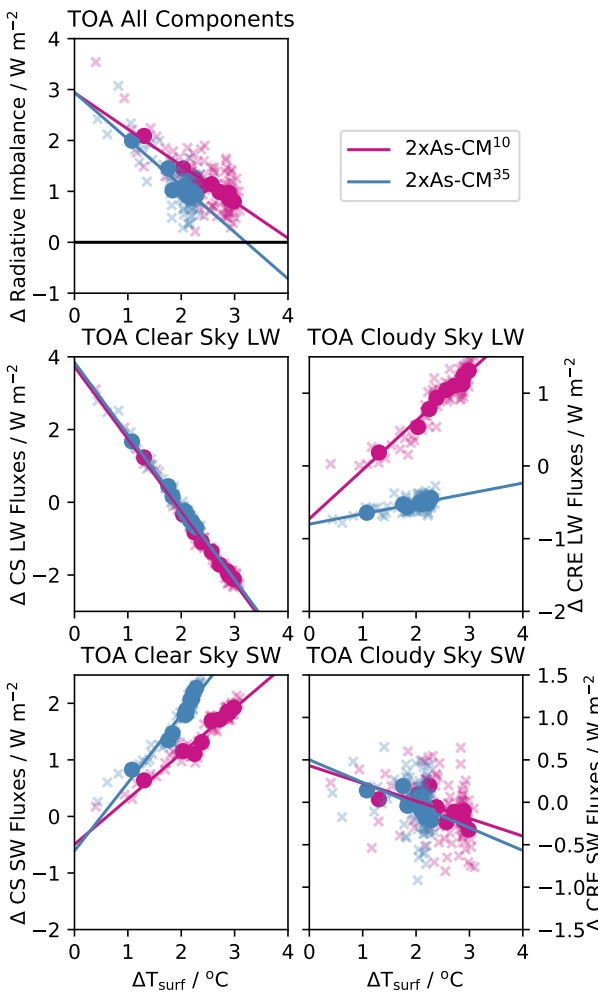

**Figure 10.** Gregory analysis of HadCM3-BL: Regression of top-of-atmosphere radiative imbalance against surface air temperature change (solid lines) for the first 100 years of 2xAs-CM[10] (pink) and 2xAs-CM[35] (blue) cases. Annual averages are indicated by crosses and decadal averages are indicated by filled circles. The regression was performed on the decadal averages.

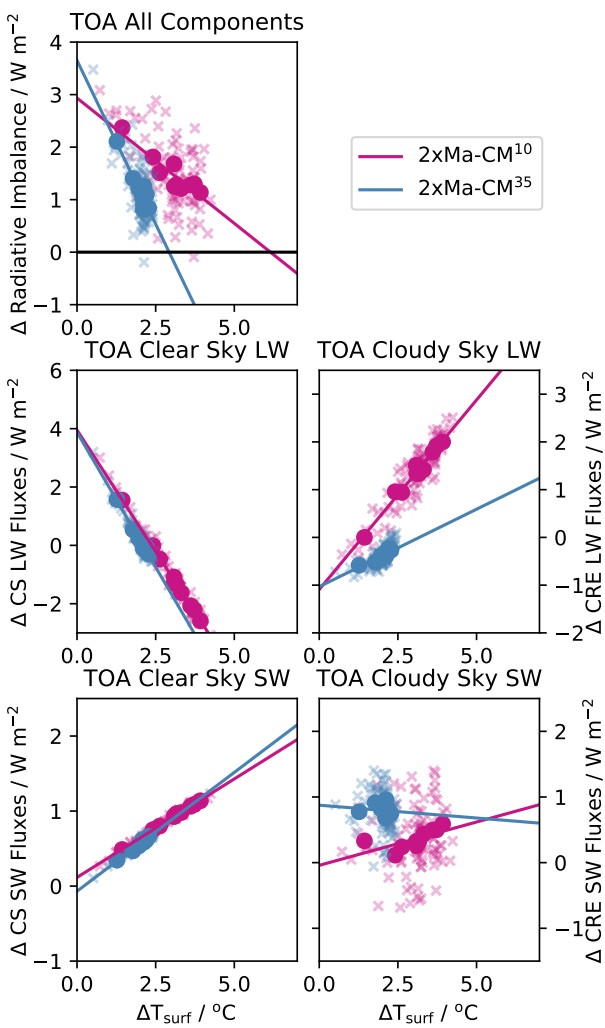

**Figure 11.** Gregory analysis of HadCM3-BL: Regression of top-of-atmosphere radiative imbalance against surface air temperature change (solid lines) for the first 100 years of 2xMa-CM[10] (pink) and 2xMa-CM[35] (blue) cases. Annual averages are indicated by crosses and decadal averages are indicated by filled circles. The regression was performed on the decadal averages.

GEM and PI-GEM in Fig. 4). For reference, HadGEM3 has a climate sensitivity of +3.6 °C (Nowack et al., 2014) while HadCM3 has a climate sensitivty of +3.1 °C (Johns et al., 2006). From the 4xPI-GEM and PI-GEM experiments, a reduction in climate sensitivity of 0.65 °C can be inferred, based on the changes in surface temperatures. For HadCM3, $CO_2$-doubling experiments were performed and a regression of the change in top-of-atmosphere radiative imbalance against change in surface

temperature following Gregory et al. 2004 (see also section 2.4) is shown in Fig. 9. The PI-CM[35] climate state has a smaller ECS than PI-CM[10] by 0.7°C. While the changes in total radiative forcing $F$ are very similar, $\xi$ is less negative (-1.08 vs -1.37 $Wm^{-2}\,°C^{-1}$) at low $pO_2$. The decomposition of these changes into their longwave and shortwave components, clear-sky and cloudy-sky components is also shown in Fig. 9. The clear-sky longwave radiative flux changes are slightly higher in 2xPI-CM[35] (4.0 $Wm^{-2}$) than 2xPI-CM[10] (3.8 $Wm^{-2}$) as would be expected due to the pressure broadening of $CO_2$. The clear

driver for the less negative $\xi$ value are from the longwave cloud radiative effect changes which is much steeper for 2xPI-CM[10] (+0.62 $Wm^{-2}\,°C^{-1}$) than 2xPI-CM[35] (+0.17 $Wm^{-2}\,°C^{-1}$). This is somewhat offset by stronger clearsky shortwave radiative feedbacks in 2xPI-CM[35] (+1.00 $Wm^{-2}\,°C^{-1}$) than 2xPI-CM[10] (+0.57 $Wm^{-2}\,°C^{-1}$). This highlights the important role that cloud radiative feedbacks play in determining the climate sensitivity.

An increase in ECS appears to be robust across the HadCM3-BL experiments. For As-CM, ECS is 0.8°C lower at 35 % $O_2$

than 10 % $O_2$ (Fig. 10). For Ma-CM, this value is much larger. A 3.3°C reduction in ECS is simulated, which is also driven by the longwave cloud radiative effects in conjunction with a weaker clear sky shortwave radiative effect which tended to cool the low $pO_2$ (Fig. 11). Unlike the clear sky shortwave effects, the longwave cloud radiative effects seem consistent across the three experiments. It should be noted that attempts were made to simulate 2x-experiments for the Wuchiapingian, however what would have been the 2xWu-CM[10], in the nomenclature used here, was numerically unstable.

The increase in climate sensitivity appears to be linked to the reduction in temperature anomaly in a warmer climate state. We propose that this is due to more vigorous convection at low $pO_2$ (Goldblatt et al., 2009) leading to an atmospheric moistening (Fig. 12) which causes warming analogously to Rose and Ferreira (2013). This is consistent with the increases in climate sensitivity observed - in a warmer climate the atmosphere can hold more water vapour, so any changes to water vapour will be amplified in their impacts on the radiative budget of the atmosphere. This water vapour feedback is also consistent with the

weaker clear sky shortwave radiative effect observed in 2xMa-CM and the temperature response observed in the Wuchiapingian simulations.

### 3.4  Response of Permian vegetation to $pO2$

Changes in $pO_2$ and surface temperatures have the potential to impact the terrestrial carbon cycle by altering the competition between the oxidative and photosynthetic metabolic pathways for Rubisco. Beerling and Berner (2000) simulated significant

changes to vegetation productivity in the Permian due to changes in oxygen content. The modelled changes to vegetation in the final 50 years of the Asselian and Wuchiapingian experiments are investigated. Focusing on changes to vegetation across the Permian, the dominant vegetation fractions for As-CM[35], As-CM[10], Wu-CM[35] and Wu-CM[10] are shown in Fig. 13. For low $pCO_2$ in the Asselian, increasing $pO_2$ leads to a reduction in the extent of broadleaf trees and greater proliferation of grasses and shrubs. This would be consistent with increases in photorespiration at high $pO_2$. The reverse is true in the Wuchiapingian

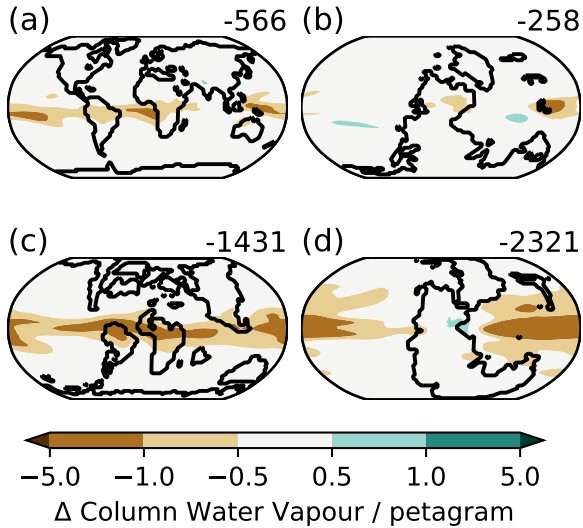

**Figure 12.** Change in column water vapour in (a) PI-CM$_{10}^{35}$, (b) As-CM$_{10}^{35}$, (c) Ma-CM$_{10}^{35}$ and (d) Wu-CM$_{10}^{35}$. Global sum values (petagrams) are offset to the top-right of each plot. Note the atmospheric drying at high $p$O2 is enhanced in the warmer climate states of the Wuchiapingian and Maastrichtian and more subdued in the cooler climate states of the Asselian and Holocene.

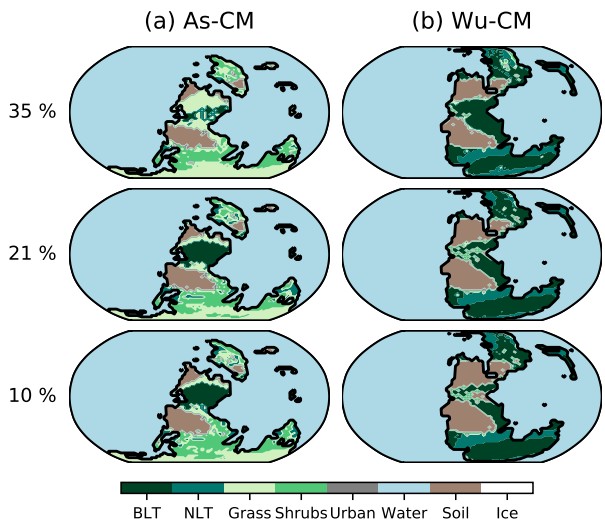

**Figure 13.** Dominant surface type for each oxygen level simulation for (a) As-CM and (b) Wu-CM. BLT: Broadleaf trees, NLT: Needleleaf trees.

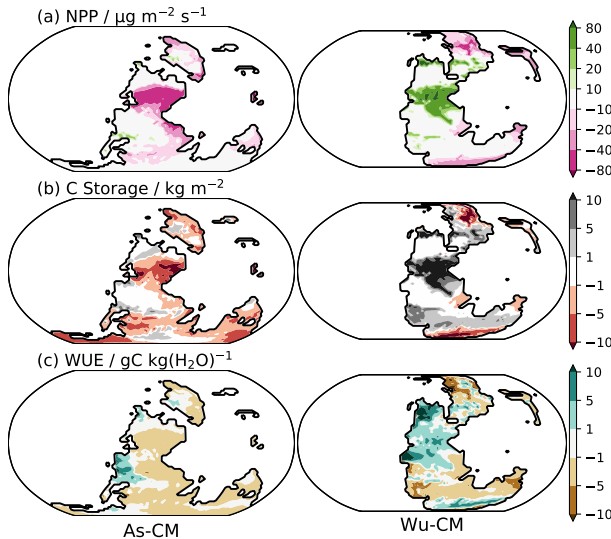

**Figure 14.** As-CM$_{10}^{35}$ (left) and Wu-CM$_{10}^{35}$ (right) anomalies for (a) net primary productivity, (b) total carbon storage and (c) water use efficiency.

simulations with increases in the extent of tropical broadleaf forests. It should be noted that the simulation of plant functional types is carefully tuned to present day vegetation which was likely considerably different in the past. Therefore, caution should be exercised when extrapolating to past vegetation changes.

Figure 14a shows the change in net primary productivity (NPP) for As-CM$_{10}^{35}$ and Wu-CM$_{10}^{35}$. The Asselian simulations
shows a large reduction in net primary productivity (NPP) as $p$O$_2$ is increased (Figure 14a, –59 Pg C yr$^{-1}$) while the reverse is true in the Wuchiapingian simulations (+33 Pg C yr$^{-1}$). At low $p$CO$_2$, it is expected that competition for Rubisco will be won out by O$_2$ and therefore that rates of photorespiration should lead to a decline in photosynthesis. This is reflected in the gross primary productivity (GPP, –34 %) and NPP (–52 %) response for the Asselian. During the Wuchiapingian, there may be sufficient CO$_2$ that competition is much less sensitive to the $p$O$_2$ so changes to NPP are much less significant. In fact, NPP is
increased by 14 % (GPP +18 %). Tropical water use efficiency is also higher in Wu-CM$^{35}$ (Figure 14c), which suggests that water economy of plants could alter to adapt to a higher $p$O$_2$ (Beerling and Berner, 2000).

These net primary productivity changes are reflected in the total carbon storage (Figure 14b) which is lower as $p$O$_2$ is increased in As-CM (–338 Pg C) and higher as $p$O$_2$ is increased in Wu-CM (+379 Pg C). This is dominated by changes in the tropics (in agreement with Beerling and Berner 2000), where broadleaf trees cover more area in Wu-CM$^{35}$. Cooler terrestrial
tropical temperatures, particularly in the warm month (Figure 4e) reduce the $p$O$_2$ inhibition of Rubisco and reduce the rate of respiration by vegetation and soils (Long, 1991; Beerling and Berner, 2000).

As these simulations are fully coupled and changes to oxygen content affect temperatures, radiation and precipitation it is challenging to explore all the possible contributions to differences between these results and the more idealised Beerling and Berner (2000) simulations. However, there is general agreement that changes occur in the signs of the response of NPP and

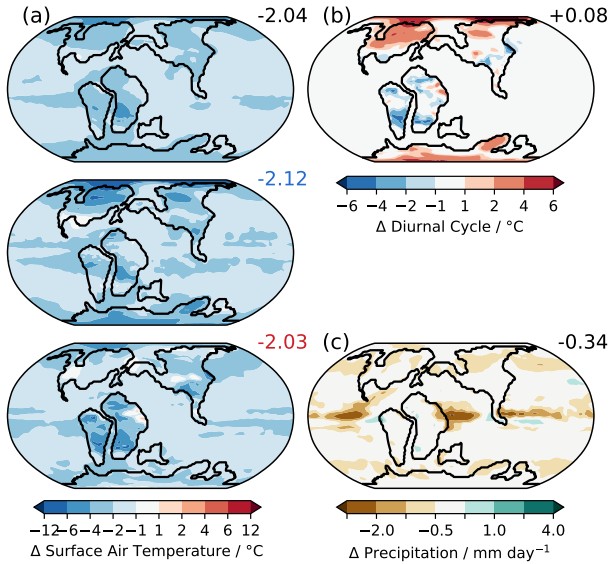

**Figure 15.** $21\% - 10\%$ O$_2$ anomalies for Poulsen et al. (2015) simulations. (a) Annual mean, cold month mean and warm month mean surface air temperature difference. (b) Change to diurnal cycle and (c) annual mean precipitation (mm day$^{-1}$).

total carbon storage. This supports the conclusions of Beerling and Berner (2000) that high $p$O$_2$ in the early Permian may have played an important role in the evolution of plants. Note that while the atmosphere and vegetation are coupled in the physical sense, the carbon cycle is not interactive (atmospheric CO2 is fixed) so determining the impacts of atmospheric $p$O2 on the carbon cycle remains an outstanding problem.

## 4   Discussion

Through its impact on atmospheric mass, oxygen content has the capacity to alter the radiative budget of the atmosphere and therefore has implications for Earth's climate. These simulations suggest that the interactions between radiative and dynamical feedbacks lead to some consistent climatic changes in HadCM3-BL with increasing $p$O$_2$:

- Reduction in the seasonal cycle in surface air temperature.

- Reduction in equator-to-pole temperature gradient.

- Reduction in global precipitation.

HadCM3-BL simulates a reduced equilibrium climate sensitivity mainly due to changes in longwave cloud feedbacks. HadGEM3-AO results also support a reduced sensitivity to CO$_2$ content at high $p$O$_2$. The pre-industrial Holocene results are supported by 1D radiative convective simulations (Payne et al., 2016), 2D model simulations (Chemke et al., 2016) and slab ocean 3D model simulations of the Archean (Charnay et al., 2013). This raises a discrepancy with the Poulsen et al. (2015) study, which

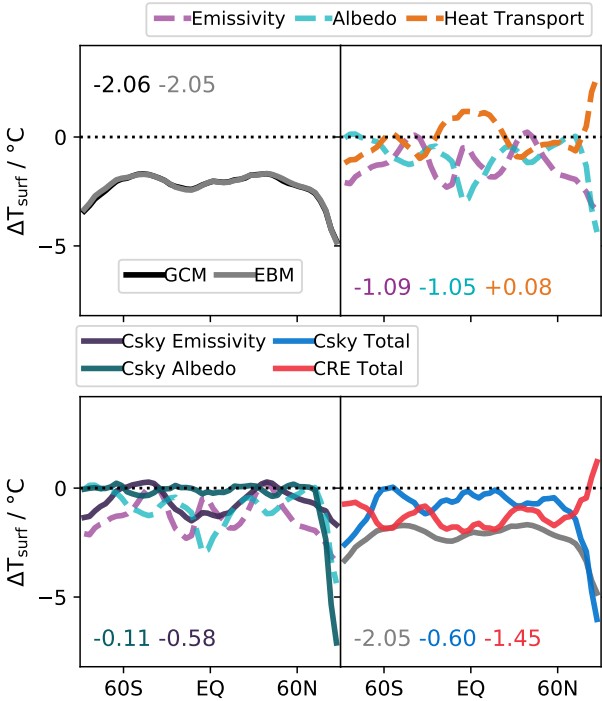

**Figure 16.** 1D-energy balance decomposition analogous to Fig. 7 for the 21–10 % $O_2$ Poulsen et al. (2015) simulations.

simulated a reduction in global mean surface temperature when increasing oxygen content in the GENESIS model. Figure 15a shows the surface air temperature change between the 10 % and 21 % Cenomanian (100.5–93.9 Ma) simulations from the Poulsen et al. (2015) study. These show a –2.04°C change in the annual mean. To understand the mechanisms behind this, we performed the 1D-energy balance decomposition on the Poulsen et al. (2015) Cenomanian 21–10 % model output. The

results are shown in Fig. 16. This shows that the cloudy-sky contribution to the temperature change dominates the climate response, contributing –1.45°C. However, the clear sky contribution is also negative (–0.60°C) including both clear-sky emissivity (–0.58°C) and clear-sky albedo (–0.11°C). This appears to support the argument that tropical cloud feedbacks explain the discrepancy between the Poulsen et al. (2015) simulations and results of 1D radiative convective models (Goldblatt, 2016), however this cannot be the only factor. An increase in pressure broadening of absorption lines would be expected to lead to

a positive contribution from the clear-sky emissivity. This suggests that cloud feedbacks alone cannot explain the discrepancy and that the implementation of pressure broadening may play a role in the anomalous Poulsen et al. (2015) response. In addition, changes to the seasonal cycle (Figure 15a) simulated by Poulsen et al. (2015) are also inconsistent with the HadGEM-AO and HadCM3-BL results, in which all simulations led to a reduced seasonal cycle as $pO_2$ increases. The Poulsen et al. (2015) Cenomanian simulations actually simulates a larger seasonal cycle at high $pO_2$ which is challenging to reconcile with the

radiative and physical processes. Note that the Poulsen et al. (2015) simulations were for an earlier Cretaceous period (Ceno-

manian) than those performed in HadCM3-BL (Maastrichtian), however the continental configurations and the global mean temperatures are reasonably similar (22.2 °C in HadCM3-BL vs 20.5 °C in Poulsen et al. (2015)).

The simulations presented in here suggest that perturbations to the wind-driven ocean circulation by increasing atmospheric mass leads to warmer temperatures, particularly at high latitudes. The magnitude of the results varies depending on the precise continental configuration and background climate state. Gyre circulations vary between the preindustrial and the Maastrichtian and Asselian case studies. Given the importance of the wind-driven ocean circulation response this suggests that a 3D representation of ocean circulation is necessary in order to capture the temperature response to atmospheric mass changes. It should be noted however that Charnay et al. (2013) simulated higher surface temperatures for the early Earth at high atmospheric mass with a slab ocean model.

The use of 3D oceans is now widespread in the palaeoclimate community, however this is not widely used in the exoplanet/early earth community (e.g. Kilic et al. 2017) and for early Earth studies such as the Archean (e.g. Charnay et al. 2013). While boundary conditions for these studies are sparse or in some cases non-existent the additional uncertainty associated with using a slab ocean should be considered. AO-GCM studies remain the best way to assess the complex coupling between potentially competing radiative and dynamical effects.

One criticism of high oxygen variability in the Phanerozoic is the possibility of runaway fire at high oxygen contents (Watson et al., 1978). While subsequent experiments have put this in doubt (Wildman et al., 2004), fire is undoubtedly a negative feedback on oxygen content. However, the cooling of warmest month temperatures over tropical and midlatitude continents in Wu-CM[35] may provide somewhat of a protective mechanism against runaway fire regimes taking hold. Lightning is a major cause of paleofire (Scott and Jones, 1994) so the reduction in convection at high $pO_2$ would also lead to fewer lightning strikes which would reduce fire initiation. In addition, higher fire risk could have favoured the evolution and spread of more fire-resistant species (Robinson, 1990).

The simulations of Permian climate (As-CM and Wu-CM) also suggest a strong role for $pO_2$ variability in the terrestrial carbon cycle. However, there are many limitations to the modelling approach employed here. The plant functional types employed here are the same as present day. In particular, $C_4$ photosynthesis likely evolved in the Oligocene (Sage, 2004) although there is evidence of vegetation which causes $C_4$-like fractionation in the Mississippian, suggesting different vegetation adaptations operating in the past (Jones, 1994). In addition, angiosperms did not evolve until the Cretaceous so gymnosperms such as cycads were more widespread in the Permian (Taylor et al., 2009). TRIFFID and other dynamic plant models were not developed with these changes in plant types in mind so simulating past vegetation changes is still a considerable challenge. However, scientific understanding of the role of plants in the climate in the Paleozoic is still immature. While early evidence suggested that late Paleozoic vegetation was unproductive based on analysis of the closest modern relatives, this perspective is increasingly being challenged (Wilson et al., 2017). Other approaches such as trait based methods (Van Bodegom et al., 2012; Porada et al., 2016) may be able to achieve more insights into the role of $pO_2$ in the Earth system. We also have not accounted for changes to the ocean carbon cycle. A biogeochemical model study suggests that pervasive oceanic anoxia and euxinia only occur below an oxygen level of around 10 % (Ozaki and Tajika, 2013) which may be below the fire threshold (Belcher et al.,

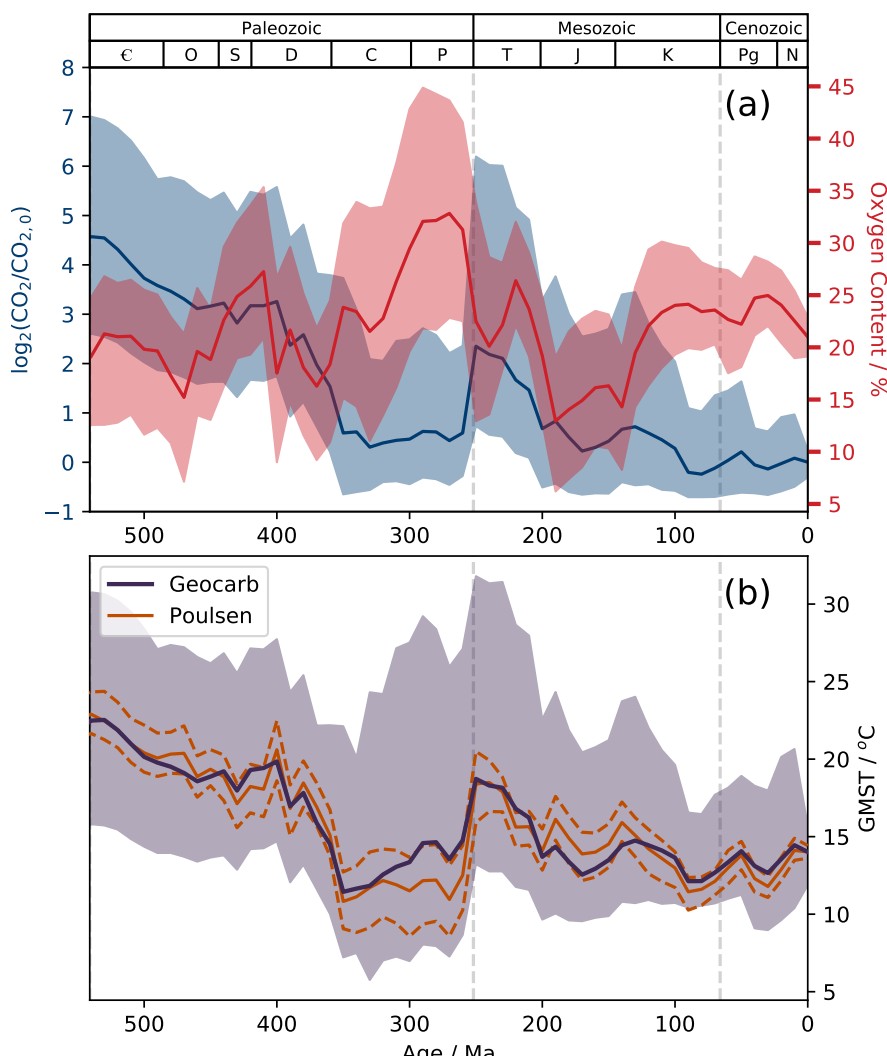

**Figure 17.** (a) Reconstructed $CO_2$ (doublings from Pleistocene values, blue) and $O_2$ content (red) and 95% confidence intervals (shading) from Royer et al. (2014) Geocarb simulations. (b) GMST reconstructed using Geocarb $pCO_2$ and climate sensitivity values (purple) and the uncertainty in GMST from $pCO_2$ uncertainty (purple shading). GMST reconstructed, accounting for $pO_2$ according to Poulsen et al. (2015) global mean temperature sensitivities (solid orange) and the uncertainty due to Geocarb $pO_2$ (orange dashed).

2010) and therefore not of relevance to many periods in the Phanerozoic. However, the extent of oceanic anoxic events may be sensitive to atmospheric $pO_2$ (Clarkson et al., 2018).

Given the small changes in global mean surface temperature (GMST, 1.5 °C maximum) compared to ECS (~3 °C), this raises the question of how much $pO_2$ variability contributes to uncertainty in Phanerozoic surface temperature even with such large uncertainties in $pO_2$ reconstructions. Figure 17 shows reconstructed Phanerozoic surface temperatures based on $CO_2$ content

and climate sensitivity from the Geocarb model (purple, Royer et al. 2014). The uncertainty associated with the 95% confidence interval in simulated $p$CO$_2$ is also indicated (purple shading). Analysis of the Poulsen et al. (2015) simulations suggests a global mean surface temperature reduction of 0.21 °C per percentage increase in O$_2$. Accounting for the $p$O$_2$ simulated by Royer et al. (2014) leads to a mean absolute difference in global mean surface temperature of 0.80 °C and maximum absolute difference of 2.59 °C (Figure 17 orange line). The largest deviations from the Geocarb values occur during the largest deviations from present atmospheric levels of O$_2$ during the Permian. However, $p$O$_2$ contributes little to the uncertainty in reconstruction of global mean surface temperature compared to $p$CO$_2$ (Figure 17 orange dashed lines), even if the temperature changes simulated by Poulsen et al. (2015) are reasonable. The HadGEM3-AO and HadCM3-BL simulations show even less sensitivity of global mean surface temperature to $p$O$_2$ changes which suggests this is likely an overestimate.

$p$O$_2$ therefore remains a secondary contribution to climatic variability in the Phanerozoic in agreement with (Payne et al., 2016) but most likely to be important during the Permian. The Artinskian (early Permian, 283.5–290.1 Ma) is associated with a rapid increase in CO$_2$ content from ~500 to ~3500 ppmv which is associated with considerable restructuring of tropical vegetation (Montañez et al., 2007). The results in this study suggest that $p$O$_2$ variability could have modulated the climate and terrestrial vegetation response to this increase in CO$_2$ content. Feulner (2017) suggested that Earth was close to entering a Snowball Earth in the late Carboniferous, when $p$O$_2$ was higher than today. We hypothesise that the carbon cycle and physical climate feedbacks described in this paper would strongly mitigate against this. If $p$CO$_2$ and $p$O$_2$ are intimately linked such that cooler climates tends to increase $p$O$_2$ this would suggest that $p$O$_2$ responses have helped to prevent Snowball Earth initiation in the Phanerozoic.

## 5  Conclusions

The numerical simulations performed in this study reconcile the surface temperature response to oxygen content changes across the hierarchy of model complexity:

- Under pre-industrial Holocene conditions, increasing atmospheric $p$O$_2$ leads to an increase in global-mean surface temperature in agreement with 1D radiative-convective model simulations. This increase is greater in the cold-month mean than the warm month-mean. The equator-to-pole temperature gradient is reduced, particularly in the cold-month mean, consistent with a stronger greenhouse effect at high atmospheric pressure.

- Lower incident surface shortwave radiation leads to a slow down of the hydrological cycle. Precipitation decreases globally under high $p$O$_2$, with regional variations.

- The climate sensitivity is lower at high $p$O$_2$, particularly in the Maastrichtian. This appears to reconcile the results of the 1D and 3D modelling approaches.

- The climate response simulated by Poulsen et al. (2015) is inconsistent with the radiative changes when considering a 1D-energy balance model decomposition of the surface temperature changes. Tropical cloud feedbacks alone were not sufficient to explain the discrepancy.

– The climate response to oxygen content variability is state-dependent so should be considered on a case-by-case basis. However, the changes are relatively small compared to the role of $CO_2$ in the Phanerozoic (Royer et al., 2014).

*Code and data availability.* Processed model output and analysis scripts will be made available on the NERC data centre. The Met Office Unified Model is available for use under licence. Please see http://www.metoffice.gov.uk/research/modelling-systems/unified-model for more
information. The ocean bathymetry and land orography reconstructions are ©Getech. Readers who would like advice on how to implement alterations to $p$O2 in their climate model are encouraged to contact the corresponding authors. UM users can obtain the code changes for these particular versions from the corresponding authors.

*Author contributions.*

The study was performed by DCW and conceived by ATA and DCW and refined with input from PJV and the co-authors.
DCW and ATA led the preparation of the manuscript and all co-authors helped in proof-reading and checking of the manuscript.

*Competing interests.* The authors declare no competing interests.

*Acknowledgements.* DCW acknowledges support from a UK Natural Environment Research Council Doctoral Training Program studentship (Ref 1502139). This work was carried out using the computational facilities of the Advanced Computing Research Centre, University of Bristol http://www.bris.ac.uk/acrc/. We acknowledge use of the MONSooN system, a collaborative facility supplied under the Joint Weather
and Climate Research Programme, a strategic partnership between the Met Office and the Natural Environment Research Council. DCW acknowledges Eric Wolff and David Stevenson for their comments on the PhD thesis from which this paper largely forms a part.

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
