# Peer review of "Simulating the Climate Response to Atmospheric Oxygen Variability in the Phanerozoic: A Focus on the Holocene, Cretaceous and Permian"

_Climate of the Past, 2018_

## Referee Comment (RC1) · Kasting (Referee) · 10 Jan 2019

This paper describes an exhaustive study of the effects of changing pO2 on Phanerozoic climates. The calculations are carried out with two different state-of-the-art climate models, one of which is an Earth system model that includes coupling between the atmosphere and the biota (forests in particular). This paper responds to a disagreement between the original 3-D climate simulations by Poulsen et al. (2015) and subsequent 1-D simulations by Goldblatt (2016) and Payne et al. (2016) (the latter of which is my own research group). The Poulsen et al. model predicted that higher pO2 leads to lower surface temperatures; the two 1-D models predicted just the opposite. The new paper basically agrees with the 1-D models, i.e., high pO2 leads to higher surface temperatures. But the results are more complicated. High pO2 can actually lead to

lower surface temperature when the starting state is a warm climate. In general, the calculations seem to be well done, and the results are well described. I have a few minor comments below. But this paper should be useful in close to its present form. The basic conclusion, which is restated in point 4 below, is that changes in $pO_2$ are secondary drivers of Phanerozoic climate. The main driver is changes in $pCO_2$, and also in solar luminosity, which is not mentioned too many times explicitly but which is implicit in the calculations.

1. (p. 3, l. 2) 'Indeed, there is support for elevated O2 by carbon isotope measurements (Beerling et al., 2002).'

–Elevated O2 during what time period?

2. (p. 9, l. 1) 'Proxy data for the Maastrichtian was obtained..'

–Proxy data..were obtained. ('Data' is plural.) This same mistake is found elsewhere.

3. (p. 19, l. 18) 'The changes in terrestrial carbon storage are equivalent to 56% of the atmospheric CO2 content in the Asselian and 16% in the Wuchiapingian which suggests that pO2 induced Earth system feedbacks could have significant impacts on atmospheric pCO2.'

–No, I don't buy this argument. Think of the numbers and the relevant time scales. Today, the atmospheric CO2 reservoir is about 1/60th the size of the dissolved inorganic carbon (DIC) reservoir in the ocean. On long time scales (> 0.5 m.y.), what changes is the total CO2 content of the combined atmosphere-ocean system. Sequestering 56% of atmospheric CO2 in forests is a trivial change to this combined reservoir. Forests can only directly affect atmospheric CO2 if they are chopped down or regrown on very short time scales, less than the time required for the atmosphere and ocean to equilibrate.

4. (p. 27, l. 11) 'pO2 therefore remains a secondary contribution to climatic variability in the Phanerozoic but most likely to be important during the Permian.'
–The first part of this statement is essentially what we said in the conclusion section of Payne et al. (2016): 'Given the large uncertainties in past levels of both O2 and CO2, we agree with Berner [2006] that Phanerozoic climate has been driven largely by changes in atmospheric CO2 and solar luminosity, coupled with changes in continental geography.' So, we are in fundamental agreement on this question.

5. (p. 27, l. 17) 'If pCO2 and pO2 are intimately linked such that cooler climates tends to increase pO2 this would suggest that pO2 responses have helped to prevent Snowball Earth initiation in the Phanerozoic.'

–Cool! I like this result. The effect of pO2 on climate is not strong, but it may help to prevent Phanerozoic Snowball Earth events.

Jim Kasting Penn State

---

## Referee Comment (RC2) · Anonymous Referee #2 · 11 Jan 2019

**Simulating the climate response to atmospheric oxygen variability in the Phanerozoic**

**Recommendation**: Accept after moderate revisions provided that the authors can adequately answer to my 2 major comments.

Anonymous

**Summary:**

Wade et al. quantify the climatic impact of changing atmospheric oxygen concentrations ($pO_2$) using two ocean-atmosphere general circulation models in the Holocene, the Cretaceous and the Permian. They systematically conduct their simulations at 3 $pO_2$ levels (10, 21 and 35 ‰), which are shown to reasonably cover the $pO_2$ changes reported during the Phanerozoic. In their model, higher $pO_2$ values (and associated greater atmospheric mass) lead to two competing effects: an increase in Rayleigh scattering that induces an increase in albedo and surface cooling, and an increase in greenhouse effect that leads to surface warming.

The authors first run the two models on the preindustrial Holocene configuration. Interestingly, the state-of-the-art IPCC-class model (HadGEM-AO) and the version of the model designed for deep-time studies (HadCM3-BL) provide climatic responses that agree at first-order, thus supporting the robustness of the subsequent deep-time HadCM3-BL integrations. In their Holocene simulations, the mean annual global climate response to an increase in $pO_2$ is a warming, with varying regional patterns. The warming is particularly strong in the northern high latitudes, especially during the cold month. A cooling is simulated at low latitudes, which is especially strong and extends to most continental areas during the warm month. Higher $pO_2$ values tend to flatten the equator-to-pole temperature gradient. They also lower the climate sensitivity to atmospheric carbon dioxide.

Then the authors run the HadCM3-BL model in the Cretaceous and two Permian time slices. They show similar climatic behaviors and discuss two specific points related to each case study: the response of the terrestrial vegetation to changing $O_2$ levels during the Permian and the impact of changing $O_2$ levels on the capacity of their model to simulate the low latitudinal temperature gradients traditionally reconstructed for the Cretaceous based on proxy data. They notably show that changing oxygen concentration only slightly improves model-data agreement in the Maastrichtian.

Last but not least, they propose a quantification of the uncertainty in global temperature resulting from uncertainties in the $pO_2$ during the Phanerozoic. They show that the temperature bias associated with poorly constrained $pO_2$ levels is significantly lower than the uncertainty associated with the lack of constraints on the $pCO_2$, with a notable contribution of $pO_2$ during the Permian though.

It should be noted that Wade et al.'s implementation of $O_2$ forcing leads to results that agree at first-order with most previous attempts, but differ in sign with the simulations of Poulsen et al. (2015; 10.1126/science.1260670). Analysis of the model runs led the authors to suggest that Poulsen et al.'s implementation may not be totally coherent.

**General comments:**

I think that Wade et al. provide a very interesting and innovative study that shades new light on the

poorly explored question of the potential impact of changing $pO_2$ levels on deep-time climate. The results are based on numerous general circulation model simulations using two generations of climate models. The manuscript is relatively well organized (an exception if the methods section, see comments) and richly illustrated with high-quality figures and abundant information embedded in tables. The manuscript is lengthy (as testified by the length of my summary above). This is essentially due to the large amount of diagnostics provided by the authors but I also suggest below deleting the section of the manuscript relative to the impact of wind stress, which I think is not very useful and relatively badly integrated in the manuscript (see hereafter).

The discussion of the discrepancy with Poulsen et al.'s (2015) results is well conducted. Indeed, Wade et al. not only compared their results with the diagnostics provided by Poulsen et al. but also downloaded and analyzed the climatic simulations of the latter, by repeating key diagnostics that they previously provided for their own model runs. This effort deserves to be acknowledged. As a reviewer of this paper, I would be happy to have Poulsen et al.'s response, be it as another review or at least as a comment on the ClimPast Discussion forum. Therefore, I encourage the Editor to contact Poulsen et al. I also encourage the authors to make their implementation of $O_2$ forcing available online (as numerical – fortran – code or as equations) in order to allow other research groups to conduct similar experiments using other climate models, thus permitting to determine to what extent the discrepancy between Poulsen et al.'s results and theirs is model-dependent (and conversely, implementation-dependent; see major comment).

Most of my comments are intended to help the authors sharpen and clarify their manuscript. The only (other) major (potentially critical) comment I have regards the robustness of the analyzed climatic simulations. I suggest accepting this manuscript with moderate revisions, provided that Wade et al. can demonstrate that their climatic simulations are robust (i.e., sufficiently close to equilibrium).

Please note that the text refers in several places to supplementary figures. I did not find any SOM.

**A. Major comments:**

- On the discrepancy with Poulsen et al.'s results. Since the current study casts doubts about the Poulsen et al. implementation of $O_2$ forcing, I suggest making the implementation of Wade et al. available online to allow other modelers to repeat such experiments using alternative climate models – using GENESIS in particular. I think that such common effort will allow improving the implementation of oxygen forcing in a collaborative and efficient way.

- On the robustness of the climatic simulations. I recently had the opportunity to attend a presentation by Dan Lunt showing that the climatic simulations published by Lunt et al. (doi:10.5194/cp-12-1181-2016), run for 1422 years, did not reach equilibrium. A longer duration in the order of 10 kyrs is necessary to reach deep-ocean equilibrium, with the global mean SST simulated at the end of the longer simulation significantly differing (several °C) from the SST simulated after 1422 years of model integration time. Therefore I logically wonder if the climatic results used in the present manuscript based on the 1422-year long model integrations (see page 6, line 15) can be trusted. To what extent are the model runs equilibrated? I encourage the authors to clarify this point. Otherwise, the subsequent publication of longer model runs may significantly question the robustness of this entire study.

  ALSO: Page 8, line 6. "iterated for 100, 1000 and 100 years". What's the justification for the 100-year integration time used for two of the 3 experiments? I doubt that such duration is sufficient to reach equilibrium under a doubled CO2 level.

**B. Other comments:**

- Title: I would suggest revising the title to clearly indicate that several case studies are considered – maybe something like: "Simulating the climate response to atmospheric oxygen variability in the Phanerozoic – Holocene, Carboniferous and Permian case studies". In my opinion, such title would be more instructive, notably permitting readers interested in these 3 key time slices to more easily find this paper.

- Page 1, Line 1. "10 %": Fig. 1 suggests that it could have reached lower values.

- Page 1, line 5. "during different climate states" > "under different…".

- Page 1, line 15. "increasing oxygen content leads to a **slightly** better agreement".

- Fig. 1. Please show the different time slices used in each case study using for instance vertical lines.

- Fig. 1 caption. "High and low limits on atmospheric oxygen are indicated by horizontal grey dashed lines". What does that mean? Please clarify. I guess those horizontal lines indicate the 2 end-member $O_2$ levels considered in the deep-time case studies. In this case, the lower line is wrongly placed in the figure (this is not 10 %).

- Page 3, line 1. "to 20–35 % in the Permian **and subsequently stabilized at levels around 15–30 % from the Mid Triassic onward**" or similar.

- Page 3, line 6. See studies by Dahl et al. (doi:10.1073/pnas.1011287107) and Lu et al. (doi: 10.1126/science.aar5372) though, which provide very interesting insights into the evolution of $p$$O_2$ during the Phanerozoic. The authors may want to refer to these studies.

- Page 3, line 12. "visible life". OK, but I'm pretty sure it refers to the ocean realm, not to terrestrial life. Similarly, I think that the most prominent change between the Precambrian and the Phanerozoic is the advent of complex forms of life **in the ocean** during the Cambrian Explosion and subsequent Ordovician radiation.

- Page 3, line 15. "**possibly** led to the Ordovician glaciation" (there are a lot of alternative hypotheses and the spatial cover and thus climate impact of the primitive Ordovician vegetation remain poorly constrained).

- Page 3, lines 17–18. "which is consistent with a long-term sensitivity of the Earth system to $CO_2$". I do not understand what the authors want to convey here, please rephrase.

- Page 3, line 21. "continuously since the late Silurian". Fig. 5 of Algeo and Ingall (doi:10.1016/j.palaeo.2007.02.029) (below) suggests that the charcoal record is more or less continuous since the latest Devonian or so.

[Figure]

- Page 3, lines 24–34. In my opinion, this paragraph is off topic or, at least, should not be included here.

- Page 4, line 12. "Cenomanian (mid Cretaceous, **~95** Ma)".

- Page 4, line 31. "Changes to the incoming solar radiation **[reference?]**".

- Section 2 "Methods & Simulations". This section should be better organized. I suggest using subsections. Here are suggestions:

  o  Page 5, line 4. "2.1. Models"

  o  Page 6, line 16. "2.2. Experiences" or "2.2. Boundary conditions"

  o  Page 9, line 1. "2.3. Data"

  o  Page 9, line 8. "2.4. 1D energy balance model"

  o  Page 9, line 26. "2.5. Climate sensitivity"

- Page 5, lines 13–14. "A fixed vegetation distribution of plant functional types is employed". Which one? A present-day one?

- Page 5, line 33. "increases in **thickness**"

- Fig. 2. Temperature unit?

- Page 6, line 5. "limited to 4 m **thick**"?

- Page 6, lines 18–19. "as it is possible to alter the model topography and bathymetry". Please delete.

- Table 1.

  o  Please explain how the experiments name is built. As it is, the reader has to figure it out himself. The use of "2x" and "4x" in particular, is not obvious. This is placed at the beginning or in the middle of the experiment name and does not refer to any $CO_2$ level but rather seems to multiply the $CO_2$ value used in the baseline runs. Please, explain all this, for instance in the caption of Table 1. Also, what's the "*"?

  o  What's the horizontal bar delimiting the 2 parts of Table 1? I guess that "baseline runs" and "sensitivity tests" may be included to refer to each part.

  o  Here and throughout (Table 2, Fig. 6, Fig. 9, Fig. 12, Fig. 16 etc.), I would prefer to see the unit in parentheses rather than with a "/": "$CO_2$ / Pa" > "$CO_2$ **(Pa)**". The use of "/" is confusing when it does not represent a ratio.

- Fig. 3. Precipitation unit? + "Continental outline is represented with the thick black line" or similar.

- Page 7, lines 2–4. "The annual average … Figs. 2 and 3." Please move these lines and figures into the results section.

- Page 8, line 4. I guess this is "$O_2$ content".

- Page 8, line 11. This sounds unlikely, see for instance Fig. 1 of Royer et al. (doi:10.1130/1052-5173(2004)014<4:CAAPDO>2.0.CO;2).

- Page 9, line 3. "heuristically". Well, this is obviously "by hand".

- Page 9, line 24. What's $\tau_{s,ebm}$ referring to?

- Page 10, line 5. Please define "CS" and "CRE".

- Page 10, lines 14–21. Here and throughout: the text is sometimes difficult to follow because the authors do not refer to figure panels. Please explicitly include "(Fig. 4b)" etc. when appropriate. Also page 13.

- Page 10, line 23. "(Figure 4 centre)" > "(Fig. 4 middle column)"

- Page 10, lines 25–27. "These could be … reduction with height". Is this effect really significant? This could be tested with a flat Earth simulation.

- Fig. 4a. Is Panama really open?

- Fig. 4. Please define the "cold month" and the "warm month".

- Table 2.

  - Missing data for 4xPI-GEM.

  - The authors may want to include data for their $EXP^{21}_{10}$ experiments in brackets next to their $EXP^{35}_{10}$ results to permit the comparison with Poulsen et al.'s results.

- Page 12, line 9. "Comparing the surface temperature **(Fig. 4a,b)**  response". Precipitation is showed in the next paragraph.

- Page 12, line 12. "air temperature  anomalies". Precipitation is showed in the next paragraph.

- Page 12, line 12. "Fig. S1". Missing supplementary figures? Please check throughout, including on page 15, line 8 + page 18, lines 7–11.

- Page 13, line 1. "representation of polar climate processes between the two models" + amplification by polar ice feedbacks.

- Page 13, line 12. Bjerkness compensation.

- Page 13, line 15. "suggests that $pO_2$ could mediate monsoon climate". Please check in the model output.

- Fig. 6 caption. "Global mean values (mm/day) are offset". Please rephrase. As it is, this suggests that values are really offset, which would be annoying. The text label is offset.

- Fig. 7. and Fig. 8:

  - Bottom left: What are the dashed lines?

  - Bottom right: Grey line is missing in the legend.

- Fig. 9, caption. "top-of-atmosphere radiative **im**balance".

- Page 18, line 11. "numerically unstable". Any idea why?

- Section 3.4.

  - I suggest changing the title for something more specific like "Response of Permian vegetation to changing $O_2$ levels" since this section really deals with the Permian case study.

  - The temperature and precipitation dependence of the dominant PFT simulated in the Permian should also be considered. To what extent are the changes in vegetation cover and type due to changes in temperature and precipitation? Changes in precipitation in the Wu-CM runs (Fig. 6e), in particular, seem to spatially correspond to the expansion of the BLT PFT (Fig. 10). I would like to see a short analysis of the environmental affinities of the main PFTs shown on the maps in Fig. 11. I suspect

that temperature and precipitation threshold values may play a more important role than changing $O_2/CO_2$ ratios.

- Fig. 11. The color map is reversed from a to b, which makes it difficult to read. Please revise.

- Page 19, line 15. "expansive". What does than mean?

- Page 20, line 8. I guess it means that the simulated changes in carbon storage on land do not impact the $pCO_2$ level? It would be good to clearly state what the authors mean by "not interactive".

- Page 20, lines 20–22. "however, it is likely … equator-to-pole temperature gradient". Please provide references to support this statement.

- Section 3.6 "Importance of Wind Stress". I get that atmospheric mass impacts wind stress, which in turn impacts the ocean circulation and the heat transport (see page 4, line 20). Unless I get it wrong, those effects are included in the coupled ocean-atmosphere simulations conducted by the authors, which is a good point. However, I do not understand why the authors test the impact of removing wind stress. In my opinion, this section is off topic and should be deleted and possibly kept for another contribution, which would simultaneously shorten the present manuscript and leave the possibility to conduct a robust analysis of the climate response (including the response of ocean dynamics, the analysis of which is essentially lacking so far). (For this reason, I did not include the minor comments relative to this section in this review).

- Fig. 12, caption. "Proxy data locations (**Upchurch et al., 2015**) are indicated".

- Fig. 14, caption. What's the unit of precipitation in panel c? Is this an annual mean?

- Page 24, line 1. "**mainly** due to".

- Page 24, lines 2–3. "The pre-industrial Holocene … of the Archean". Please support this statement with appropriate references.

- Page 24, lines 13–15. So, the implementation of pressure broadening is not the same as in Poulsen et al.? Page 7 line 2 suggests that $O_2$ forcing is analogous to Poulsen et al. Please clearly state what's common between both studies and what's different. Also, I encourage the authors to make their numerical code available for future work (see major comment).

- Page 24, lines 18–21. Since sub-daily model output was not written on disk in Wade et al. model runs and thus not made available for analysis, I suggest deleting this comparison that is not that instructive.

- Page 24, lines 22–24. Please be cautious: even in a slab model, the continental configuration can impact the ocean heat transport due to the varying ocean area and global climate and thus temperature (which is also impacted by the continental configuration). Another, maybe more robust argument to support the comparison, is that (i) both reconstructions are not so different at first-order and (ii) both simulations provide a relatively close global climate state (compare the mean annual SAT in Poulsen et al. 21% and this study 21% – ca. 18°C vs. 22°C).

- Page 24, lines 25–31. The contribution of changing ocean dynamics / deep circulation to the simulated climate changes is addressed for the first time here. I suggest either deleting this unsupported statements or providing clear diagnostics of the changes in ocean dynamics.

- Page 25, lines 1–2. The authors may want to refer to Pohl et al. (doi:10.5194/cp-10-2053-2014), who demonstrated the importance of ocean dynamics to simulate Ordovivician climate changes.

- Page 25, lines 5–6. "Increases … high latitudes". Please provide a reference.

- Page 25, lines 3–14. The authors may also want to refer to the climatic mechanism demonstrated by Rose and Ferreira (doi: 10.1175/JCLI-D-11-00547.1), which was subsequently invoked by Ladant and Donnadieu (doi:10.1038/ncomms12771) to explain the climate changes observed in their Cretaceous model runs.

- Page 25, line 16. "While subsequent experiments have put this in doubt". Please support this statement with a reference.

- Page 25, lines 24–25. "although there is evidence of vegetation which causes C4-like fractionation". During which geological period?

- Page 25, line 31. "Other approaches such as trait based methods". The authors may want to cite Porada et al. (doi: 10.1038/ncomms12113) who applied a trait-based model to simulate the impact of the Ordovician primitive terrestrial vegetation on weathering.

- Page 25, line 32 to page 26, line 2. I suggest deleting this paragraph.

- Figure 16. The authors previously demonstrated that Poulsen et al.'s implementation of $O_2$ forcing may not be robust. I think this is thus relatively unexpected that they here use those results in their Phanerozoic calculations, even if this may constitute a conservative estimate. I suggest using the results of the current study instead.

- Page 27, lines 17–18. "If $pCO_2$ and $pO_2$ … in the Phanerozoic". Why? Why would cooler climates be associated with higher $pO_2$ levels? I cannot imagine any clear and straightforward explanation to this.

**C. Minor points:**

- Page 3, line 2. "(grey shading **in Fig.** 1)".

- Page 5, line 13. "which simulate**s**".

- Page 6, line 12. Reference formatting: "(Valdes et al., 2017)".

- Page 6, lines 20–21. Reference formatting.

- Page 7, line 3. Please use correct experiment names instead of 4 x PI-CM[21].

- Page 8, line 17. "monotically increasing ozone column". Please rephrase.

- Page 8, line 29. Reference formatting.

- Page 10, line 19. "**Wu**-CM" (lower-case).

- Page 10, line 21. "This suggests that … but is non-linear". Please revise.

- Page 12, line 14. ", which **are** strongest".

- Fig. 5, caption. "4XPI-GEM(red)": missing space. See also multiple occurrences on page 18, lines 1–5.

- Fig. 6 color bar labels. Font size issue leading to overlapping text.

- Page 19, line 6. "**Fig. 11a**".

- Page 19, line 16. "reduces" > "reduce" (2 occurrences on the same line).

- Page 20, line 13. Please delete question mark.

- Page 20, lines 25–26. "These show **that** across both $CO_2$ contents **that** increasing". Please rephrase.

- Page 22, lines 4–5. "has the capacity to alter the radiative budget of the atmosphere and therefore  Earth's climate".

- Page 22, line 6. "with increasing $p\text{O}_2$**:**".

- Fig. 15, caption. Missing space.

- Page 25, line 3. " contribute".

- Page 25, line 9. "compared".

- Page 25, line 13. "which may increase lead to". Please revise.

- Page 27, line 7. "PAL". Please write in full or provide meaning.

- Page 27, line 16. "When $p\text{O}_2$ **was** higher".

---

## Referee Comment (RC3) · Anonymous Referee #3 · 3 Feb 2019

The manuscript "Simulating the Climate Response to Atmospheric Oxygen Variability in the Phanerozoic" by Wade et al. presents results from two ocean-atmosphere global circulation models to test the response of temperature, precipitation, and climate sensitivity to variable oxygen levels in earth's past. The primary results are that increasing oxygen levels causes global temperature to increase, precipitation to decrease, and climate sensitivity to change slightly. These results lead the authors to conclude that oxygen is a secondary factor (to CO2, though presumably also to solar luminosity and paleogeography) in earth's climate history. The study is mostly very well done, interesting, and well presented. My comments are mostly minor, and should not impede the eventual publication of the manuscript in Climates of the Past.

The use of two climate models is a strength of this paper, and I commend the authors

for the extra effort. However, without a more in-depth discussion of how the models are different and how the differences lead to the responses reported in the paper, the effort falls a little short. It is worth noting and discussing that both models Edwards and Slingo (1996) radiation scheme. What about other physics schemes? Would other non-Hadley models that don't share the same physical parameterizations be expected to have larger differences than these two models?

One of the most interesting results in the study is the difference in response with geography, and specifically the fact that the Wuchiapingian simulations show a temperature response that is opposite of the other runs. This is especially interesting in light of the conflicting results from previous models. The authors need to include an analysis and explanation of this result.

The manuscript tries to do too much. Section 3.4 is one example (3.5 and 3.6 are others). The discussion of the earth system feedbacks is interesting, but I would have preferred to see it in a standalone study that could do it justice and allow for a fuller discussion of the results and limitations. One shortcoming that the authors do not address is the physiological response of plants to changes in $CO_2$ and $O_2$. How the model handles these changes needs to be described. How well do we know how plants today and in the past responded to changes in atmospheric composition? Recent literature also indicates that changes in soil respiration may be as important as changes in plant respiration. How is this handled in the model?

Section 3.5 and the discussion of other mechanisms for producing warm climates is really a distraction from the main focus of the paper. The model-data comparison is not particularly rigorous and not necessary, and the discussion of warming mechanisms is incomplete and doesn't reference many important studies. Both sections should be deleted.

Section 3.6 on the influence of wind stress is interesting, but not very insightful without a proper analysis of the explanation for the differences between runs. This section

should be removed or (preferably) expanded. How does the total heat transport differ between these runs with and without wind stress?

One of the main results of the paper is that the response to changes in O2 is very much a function of cloud feedbacks (e.g. Section 3.2). How robust then are the results? How do cloud feedbacks in HadCM and Had GEM3 compare to each other and to other models? This major point is not discussed in the Discussion or presented in the Conclusions.

P. 3, L. 17. "which is consistent with the long-term sensitivity of the Earth system to CO2 changes..." I don't understand this comment. The fact that the CO2 range is constrained should not have an influence on the climate system sensitivity to CO2.

P. 16, L. 6-7. Please state the climate sensitivity of HadGEM3-AO and HadCM3-BL.

P. 18, L. 1. "The clear-sky longwave radiative flux changes are higher in PI2X-CM..." That's not what I see in Fig. 9a. Is there a typo here, or am I misinterpreting something?

P. 18, L. 8. "For Ma-CM, this value is much larger." This is an interesting result that is not intuitive. The authors should provide a fuller explanation of the large change in sensitivity with this paleogeography and include the figure in the main text.

―――――――――――――――――――――

---

## Author Response (AR1)

**Replies to Reviewer Comments**

**David Wade on behalf of the authors**

We would like to extend our gratitude to Jim Kasting and the anonymous reviewers for the time and care they took in reviewing the paper. The comments will be dealt with in turn and changes to the updated manuscript will be described. In all cases, reviewer comments can be identified by the red text and the author reply in the black text. We've provided a tracked changes document for the editor where the new additions (bold black text here) are added in blue and text we have deleted is scored out in red for clarity.
* * *
**Jim Kasting**

This paper describes an exhaustive study of the effects of changing pO2 on Phanerozoic climates. The calculations are carried out with two different state-of-the-art climate models, one of which is an Earth system model that includes coupling between the atmosphere and the biota (forests in particular). This paper responds to a disagreement between the original 3-D climate simulations by Poulsen et al. (2015) and subsequent 1-D simulations by Goldblatt (2016) and Payne et al. (2016) (the latter of which is my own research group). The Poulsen et al. model predicted that higher pO2 leads to lower surface temperatures; the two 1-D models predicted just the opposite. The new paper basically agrees with the 1-D models, i.e., high pO2 leads to higher surface temperatures. But the results are more complicated. High pO2 can actually lead to lower surface temperature when the starting state is a warm climate. In general, the calculations seem to be well done, and the results are well described. I have a few minor comments below. But this paper should be useful in close to its present form. The basic conclusion, which is restated in point 4 below, is that changes in pO2 are secondary drivers of Phanerozoic climate. The main driver is changes in pCO2, and also in solar luminosity, which is not mentioned too many times explicitly but which is implicit in the calculations.

1. (p. 3, l. 2) 'Indeed, there is support for elevated O2 by carbon isotope measurements (Beerling et al., 2002).'–Elevated O2 during what time period?
We thank Prof. Kasting and have added the following to clarify: "... isotope measurements **during the Permian** (Berling et al., 2002)."

2. (p. 9, l. 1) 'Proxy data for the Maastrichtian was obtained..'
–Proxy data..were obtained. ('Data' is plural.) This same mistake is found elsewhere.
Thanks for spotting this. We have changed this in the text on P9 line 1, P9 line 3, P24 line 20.

3. (p. 19, l. 18) 'The changes in terrestrial carbon storage are equivalent to 56% of the atmospheric CO2 content in the Asselian and 16% in the Wuchiapingian which suggests that pO2 induced Earth system feedbacks could have significant impacts on

atmospheric pCO2.'

–No, I don't buy this argument. Think of the numbers and the relevant time scales. Today, the atmospheric CO2 reservoir is about 1/60th the size of the dissolved inorganic carbon (DIC) reservoir in the ocean. On long time scales (> 0.5 m.y.), what changes is the total CO2 content of the combined atmosphere-ocean system. Sequestering 56% of atmospheric CO2 in forests is a trivial change to this combined reservoir. Forests can only directly affect atmospheric CO2 if they are chopped down or regrown on very short time scales, less than the time required for the atmosphere and ocean to equilibrate.

We take Prof. Kasting's point here. In our study we neglected to simulate ocean biogeochemistry and as a result there are issues in interpreting the effects of changes in pO2 on the carbon cycle. In light of this and the other comments from the reviewers we suggest to leave the section on Earth system feedbacks in (section 3.4) but with some modifications to the text. As a result we have modified the text and deleted lines 18 and 19 on P19 and lines 1 and 2 on P 20 and added the following (in bold):

"Note that while the atmosphere and vegetation are coupled in the physical sense, **the carbon cycle is not interactive so determining the impacts of atmospheric pO2 on the carbon cycle remains an outstanding problem.**"

4. (p. 27, l. 11) 'pO2 therefore remains a secondary contribution to climatic variability in the Phanerozoic but most likely to be important during the Permian.'

–The first part of this statement is essentially what we said in the conclusion section of Payne et al. (2016): 'Given the large uncertainties in past levels of both O2 and CO2, we agree with Berner [2006] that Phanerozoic climate has been driven largely by changes in atmospheric CO2 and solar luminosity, coupled with changes in continental geography.' So, we are in fundamental agreement on this question.

We thank Prof. Kasting for the comment and have added the following text to clarify this point:

"pO2 therefore remains a secondary contribution to climatic variability in the Phanerozoic **in agreement with Payne et al. (2016)** but most likely to be important during the Permian."

5. (p. 27, l. 17) 'If pCO2 and pO2 are intimately linked such that cooler climates tends to increase pO2 this would suggest that pO2 responses have helped to prevent Snowball Earth initiation in the Phanerozoic.'

–Cool! I like this result. The effect of pO2 on climate is not strong, but it may help to prevent Phanerozoic Snowball Earth events.

Thank you for the comments, we agree this is indeed an interesting result.

Jim Kasting Penn State

**Anonymous Reviewer #2**

Wade et al. quantify the climatic impact of changing atmospheric oxygen concentrations (pO2) using two ocean-atmosphere general circulation models in the Holocene, the Cretaceous and the Permian. They systematically conduct their simulations at 3 pO2 levels (10, 21 and 35 ‰), which are shown to reasonably cover the pO2 changes reported during the Phanerozoic. In their model, higher pO2 values (and associated greater atmospheric mass) lead to two competing effects: an increase in Rayleigh scattering that induces an increase in albedo and surface cooling, and an increase in greenhouse effect that leads to surface warming. The authors first run the two models on the preindustrial Holocene configuration. Interestingly, the state-of-the-art IPCC-class model (HadGEM-AO) and the version of the model designed for deep-time studies (HadCM3-BL) provide climatic responses that agree at first-order, thus supporting the robustness of the subsequent deep-time HadCM3-BL integrations. In their Holocene simulations, the mean annual global climate response to an increase in pO2 is a warming, with varying regional patterns. The warming is particularly strong in the northern high latitudes, especially during the cold month. A cooling is simulated at low latitudes, which is especially strong and extends to most continental areas during the warm month. Higher pO2 values tend to flatten the equator-to-pole temperature gradient. They also lower the climate sensitivity to atmospheric carbon dioxide. Then the authors run the HadCM3-BL model in the Cretaceous and two Permian time slices. They show similar climatic behaviors and discuss two specific points related to each case study: the response of the terrestrial vegetation to changing O2 levels during the Permian and the impact of changing O2 levels on the capacity of their model to simulate the low latitudinal temperature gradients traditionally reconstructed for the Cretaceous based on proxy data. They notably show that changing oxygen concentration only slightly improves model-data agreement in the Maastrichtian.

Last but not least, they propose a quantification of the uncertainty in global temperature resulting from uncertainties in the pO2 during the Phanerozoic. They show that the temperature bias associated with poorly constrained pO2 levels is significantly lower than the uncertainty associated with the lack of constraints on the pCO2, with a notable contribution of pO2 during the Permian though.

It should be noted that Wade et al.'s implementation of O2 forcing leads to results that agree at firstorder with most previous attempts, but differ in sign with the simulations of Poulsen et al. (2015; 10.1126/science.1260670). Analysis of the model runs led the authors to suggest that Poulsen et al.'s implementation may not be totally coherent.

We would like to thank the reviewer for their time in providing such an in depth review of our manuscript and for their comments and suggestions on how to improve it.

I think that Wade et al. provide a very interesting and innovative study that shades new light on the poorly explored question of the potential impact of changing pO2 levels on deep-time climate. The results are based on numerous general circulation model simulations using two generations of climate models. The manuscript is relatively well organized (an exception if the methods section, see comments) and richly illustrated with high-quality figures and abundant information embedded in tables. The manuscript is lengthy (as testified by the length of my summary above). This is essentially due to the large amount of diagnostics

David Wade on behalf of the authors

provided by the authors but I also suggest below deleting the section of the manuscript relative to the impact of wind stress, which I think is not very useful and relatively badly integrated in the manuscript (see hereafter).
We thank the reviewer for this comment. This is also suggested by reviewer #3 and we agree that removing this from the paper will help make the overall messages clearer.

The discussion of the discrepancy with Poulsen et al.'s (2015) results is well conducted. Indeed, Wade et al. not only compared their results with the diagnostics provided by Poulsen et al. but also downloaded and analyzed the climatic simulations of the latter, by repeating key diagnostics that they previously provided for their own model runs. This effort deserves to be acknowledged. As a reviewer of this paper, I would be happy to have Poulsen et al.'s response, be it as another review or at least as a comment on the ClimPast Discussion forum. Therefore, I encourage the Editor to contact Poulsen et al. I also encourage the authors to make their implementation of O2 forcing available online (as numerical – fortran – code or as equations) in order to allow other research groups to conduct similar experiments using other climate models, thus permitting to determine to what extent the discrepancy between Poulsen et al.'s results and theirs is model-dependent (and conversely, implementation-dependent; see major comment).
We thank the reviewer for the comments and suggestions for us to provide some numerical code or equations to help others. We did consider this at length but after consideration we feel that this would end up not being useful beyond users of the specific climate models we have used in this study. We will make clear in the manuscript that users of the versions of the climate models we have used are welcome to our code changes, upon request, but these changes are so specific to our models that they would not prove useful to other modelling groups. Indeed, implementing the changes in HadGEM3-AO was very different to implementing the changes in HadCM3-BL.

Added to the *Code and data availability* section:
"Readers who would like advice on how to implement alterations to $p$O$_2$ in their climate model are encouraged to contact the corresponding authors. UM users can obtain the code changes for these particular versions from the corresponding authors."

Most of my comments are intended to help the authors sharpen and clarify their manuscript. The only (other) major (potentially critical) comment I have regards the robustness of the analyzed climatic simulations. I suggest accepting this manuscript with moderate revisions, provided that Wade et al. can demonstrate that their climatic simulations are robust (i.e., sufficiently close to equilibrium).

Please note that the text refers in several places to supplementary figures. I did not find any SOM.
We thank reviewer #2 for pointing this out and this was an error at the submission stage we will rectify in the response.

A. Major comments:
• On the discrepancy with Poulsen et al.'s results. Since the current study casts doubts about the Poulsen et al. implementation of O2 forcing, I suggest making the implementation of Wade et al. available online to allow other modelers to repeat such experiments using

alternative climate models – using GENESIS in particular. I think that such common effort will

allow improving the implementation of oxygen forcing in a collaborative and efficient way.
As we have alluded to above we don't feel that this would be a practical action going forward. Instead, we would be very willing to help others with implementing the code changes required to repeat the calculations we have performed in their own models. Indeed, the formulation of equations used in different climate models may make it deeply difficult, if not nigh impossible, to perform these types of simulations. As we mentioned above, the code changes we made in the two different versions of the Hadley centre climate model were drastically different.

• On the robustness of the climatic simulations. I recently had the opportunity to attend a presentation by Dan Lunt showing that the climatic simulations published by Lunt et al. (doi:10.5194/cp-12-1181-2016), run for 1422 years, did not reach equilibrium. A longer duration in the order of 10 kyrs is necessary to reach deep-ocean equilibrium, with the global mean SST simulated at the end of the longer simulation significantly differing (several °C) from the SST simulated after 1422 years of model integration time. Therefore I logically wonder if the climatic results used in the present manuscript based on the 1422-year long model integrations (see page 6, line 15) can be trusted. To what extent are the model runs equilibrated? I encourage the authors to clarify this point. Otherwise, the subsequent publication of longer model runs may significantly question the robustness of this entire study.
We thank the reviewer for this comment, which touches on an important issue in climate modelling more widely. We reject the idea that the simulations we have shown are not sufficiently well spun up to allow us to make robust conclusions. The desirable length of a particular climate model integration depends intrinsically on the question being asked and on a number of factors including:

- Model components (e.g. inclusion of land ice would lead to a longer integration time due to the response time of that physical system)
- Magnitude of the imposed changes (with respect to the baseline state)
- Pragmatics (time and computational resources required to perform the integration)

The main question we ask here is how does the climate simulated by the models we've investigated, and by proxy the Earth, respond to changes in the amount of $O_2$ in the atmosphere. This is a form of a forcing-feedback study in which the forcing is being imposed by changing $pO_2$ in our climate models. Forcing-feedback studies are widely used for example to understand how climate change will evolve over the coming century. Ultimately we are trying to understand if the signal from the forcing is large or is within the internal noise present in the chaotic climate system. We believe we have robustly shown the amount of climate change from changing the amount of $pO_2$ in the atmosphere is second order compared to changes in $pCO_2$ but is not insignificant and we believe we have identified the major mechanisms behind the changes in climate. However, we wish to further elaborate on why we feel these simulations are robust. Hereafter e will refer to the doi:10.5194/cp-12-1181-2016 study as Lunt et al. 2016.

*Model Components*
While the deep ocean does take a considerable amount of time to spin-up, the interest of our study is much more in the shallow ocean response to the imposed changes. Indeed, one

simulation passing some ocean bifurcation and entering into a new pattern of circulation after several thousand model years would be an intriguing result but would not be insightful for our understanding of the impact of atmospheric $pO_2$ changes. It is worth noting that on long enough time periods, orbital changes would need to be accounted for as these are not fixed on timescales of 10 kyr so would ideally need to be integrated over a number of Milankovitch cycles in order to be properly spun up. This would be too computationally expensive for all but Earth system models of intermediate complexity, which would not be suitable for this study due to their simplistic treatment of atmospheric radiation.

*Magnitude*
The smaller the imposed change, the less time is required for integration. It is worth noting that the Lunt et al. (2016) experiment is initialised with an ocean at rest and an idealised temperature profile while the imposed atmospheric change is a quadrupling of $CO_2$ from preindustrial conditions. With no initial ocean circulation, it will take a significant amount of time for the deep ocean to respond. This kind of spin-up process is required in the context of that study, where the new continental configuration has been imposed. In our proposed study we use existing, well spun-up simulations and perform perturbations to the $pO_2$ content off these which, as shown by the model results, are substantially smaller than the impact of quadrupling the atmospheric $pCO_2$ as performed in Lunt et al. 2016.

*Pragmatics*
Given the range of time periods over which $pO_2$ has varied, it is desirable to attempt to simulate a few different time periods. In addition, the desire to perform the idealised Holocene run with a more "state-of-the-art" model means that each simulation with HadGEM3-AO took at least 120 days of continuous simulation, compared to each HadCM3-BL which run around a month. Significantly longer simulations would not be possible given the available computational resource.

We therefore believe that the simulations presented are sound, given their application to understanding the impacts of $pO_2$ variability.

ALSO: Page 8, line 6. "iterated for 100, 1000 and 100 years". What's the justification for the 100-year integration time used for two of the 3 experiments? I doubt that such duration is sufficient to reach equilibrium under a doubled CO2 level.

The 100 year simulations were designed explicitly for performing Gregory analysis to understand the forcing-feedback relationships and this requires fairly short runs. The 1000 year simulations for Ma-CM were run to enable a model-data comparison (section 3.5) which, based on the reviewers comments, we propose to remove. On this basis we will modify the text in the manuscript to read:

"The PI2x-CM*, Ma2x-CM* and As2x-CM* experiments were spun off from the end of the PI-CM, Ma-CM and As-CM experiments and iterated for **100 years in order to perform a Gregory (2004) analysis.**"

B. Other comments:
• Title: I would suggest revising the title to clearly indicate that several case studies are considered – maybe something like: "Simulating the climate response to atmospheric oxygen
variability in the Phanerozoic – Holocene, Carboniferous and Permian case studies". In my

opinion, such title would be more instructive, notably permitting readers interested in these 3 key time slices to more easily find this paper.

We thank the reviewer for this useful comment to help improve the visibility of the paper and suggest a subtle revision of the title (in bold) to:

"Simulating the climate response to atmospheric oxygen variability in the Phanerozoic: **A focus on the Holocene, Cretaceous and Permian."**

• Page 1, Line 1. "10 %": Fig. 1 suggests that it could have reached lower values.
Noted and changed in the abstract to reflect this.

• Page 1, line 5. "during different climate states" > "under different...".
Noted and changed in the abstract to reflect this.

• Page 1, line 15. "increasing oxygen content leads to a slightly better agreement".
As we are removing section 3.5 we have changed the abstract to read:
"Case studies from past climates are investigated using HadCM3-BL which show that in the warmest climate states **in the Maastrichtian (72.1-66.0 Ma)**, increasing oxygen may lead to a temperature decrease, as the equilibrium climate sensitivity is lower."

• Fig. 1. Please show the different time slices used in each case study using for instance vertical lines.
This has been added to the adjusted figure (repeated below the next comment). New text has been added to the caption: "Timings of the palaeo case studies explored in this study are indicated by the vertical dotted lines (As: Asselian, Wu: Wuchiapingian, Ma: Maastrichtian)."

• Fig. 1 caption. "High and low limits on atmospheric oxygen are indicated by horizontal grey dashed lines". What does that mean? Please clarify. I guess those horizontal lines indicate the 2 end-member O2 levels considered in the deep-time case studies. In this case, the lower line is wrongly placed in the figure (this is not 10 %).
To reduce ambiguity we have removed the grey dashed lines from the revised figure which is provided below:

[Figure]

David Wade on behalf of the authors

• Page 3, line 1. "to 20–35 % in the Permian and subsequently stabilized at levels around 15–30 % from the Mid Triassic onward" or similar.

Thanks for the suggestion which we have adopted directly in the text.

• Page 3, line 6. See studies by Dahl et al. (doi:10.1073/pnas.1011287107) and Lu et al. (doi: 10.1126/science.aar5372) though, which provide very interesting insights into the evolution of pO2 during the Phanerozoic. The authors may want to refer to these studies.

We'd like to thank the author for bringing these to our attention. We have adapted the sentence in question to "At the time of writing, there are no direct geochemical proxies for **atmospheric** $p$O$_2$ on the Phanerozoic timescale. However, there is isotopic evidence of oceanic oxygenation in steps at approximately 560 (Dahl et al 2010), 400 (Dahl et al 2010, Lu et al 2018) and 200 Mya (Lu et al 2018)."

• Page 3, line 12. "visible life". OK, but I'm pretty sure it refers to the ocean realm, not to terrestrial life. Similarly, I think that the most prominent change between the Precambrian and the Phanerozoic is the advent of complex forms of life in the ocean during the Cambrian Explosion and subsequent Ordovician radiation.

We don't mean to diminish the role of the ocean and so suggest modifying the manuscript to read:

"visible life and **one of** the marked.. ".

• Page 3, line 15. "possibly led to the Ordovician glaciation" (there are a lot of alternative hypotheses and the spatial cover and thus climate impact of the primitive Ordovician vegetation remain poorly constrained).

We have added "**possibly**" as the reviewer suggests.

• Page 3, lines 17–18. "which is consistent with a long-term sensitivity of the Earth system to CO2". I do not understand what the authors want to convey here, please rephrase.

We have removed this last part of the sentence starting on line 16 page 3.

• Page 3, line 21. "continuously since the late Silurian". Fig. 5 of Algeo and Ingall (doi:10.1016/j.palaeo.2007.02.029) (below) suggests that the charcoal record is more or less continuous since the latest Devonian or so.

We thank the reviewer for the comment and have modified the text to reflect the point:

"Charcoal appears in the fossil record continuously since the **late Devonian (~ 360 Ma, Algeo et al 2007**, Scott and Glasspool 2006)."

[Figure]

• Page 3, lines 24–34. In my opinion, this paragraph is off topic or, at least, should not be included here.
We acknowledge the reviewers comment but would rather keep this paragraph in for completeness as it helps provide some motivation for discussions on Earth system feedbacks which many readers may be interested in.

• Page 4, line 12. "Cenomanian (mid Cretaceous, ~95 Ma)".
A typo, thanks for spotting.

• Page 4, line 31. "Changes to the incoming solar radiation [reference?]".
Reference to Lunt et al (2016) added.

• Section 2 "Methods & Simulations". This section should be better organized. I suggest using subsections. Here are suggestions:
o Page 5, line 4. "2.1. Models"
o Page 6, line 16. "2.2. Experiences" or "2.2. Boundary conditions"
o Page 9, line 1. "2.3. Data"
o Page 9, line 8. "2.4. 1D energy balance model"
o Page 9, line 26. "2.5. Climate sensitivity"
We thank the reviewer for their suggestions in improving the layout of this section and have added in the subsections and headings as they have suggested.

• Page 5, lines 13–14. "A fixed vegetation distribution of plant functional types is employed". Which one? A present-day one?
We have changed the text to make it clearer:
"A fixed **present-day** vegetation distribution of plant functional types is employed."

• Page 5, line 33. "increases in thickness"
We have changed the text to:
"In the vertical, 31 model levels are used which increase **in thickness** steadily between"

• Fig. 2. Temperature unit?

We are not sure what the reviewers comment is? The figures clearly show the surface temperature in degrees Celsius -- the most commonly used unit for the variable -- indicated in both the figure and the figure caption.

• Page 6, line 5. "limited to 4 m thick"?

Changed.

• Page 6, lines 18–19. "as it is possible to alter the model topography and bathymetry". Please delete.

Deleted.

• Table 1.
o Please explain how the experiments name is built. As it is, the reader has to figure it out himself. The use of "2x" and "4x" in particular, is not obvious. This is placed at the beginning or in the middle of the experiment name and does not refer to any CO2 level but rather seems to multiply the CO2 value used in the baseline runs. Please, explain all this, for instance in the caption of Table 1. Also, what's the "*"?
o What's the horizontal bar delimiting the 2 parts of Table 1? I guess that "baseline runs" and "sensitivity tests" may be included to refer to each part.

The simulation names have been harmonised throughout such that XX-YYY refers to time period XX and model YYY, while a prefix of Nx indicates that the $CO_2$ content has been multiplied by a factor of N with respect to experiment XX-YYY, i.e. 4xPI reads "Four-times preindustrial" which is more intuitive. The horizontal lines now separate the equilibrium experiments from the transient Gregory experiments. The table caption has been expanded to read:

"Experiment names AA-BBB include the continental configuration (AA) and model used (BBB). Experiment names NxAA-BBB indicate a multiplier of $CO_2$ with respect to AA-BBB. A star (*) indicates that the $CO_2$ multiplier was applied instantaneously and the transient adjustment to climate was analysed for the purpose of a Gregory 2004  analysis."

o Here and throughout (Table 2, Fig. 6, Fig. 9, Fig. 12, Fig. 16 etc.), I would prefer to see the unit in parentheses rather than with a "/": "CO2 / Pa" > "CO2 (Pa)". The use of "/" is confusing when it does not represent a ratio.
https://www.bipm.org/en/publications/si-brochure/section5-3.html suggests that SI recommendations are to use "/" to express units so this convention has been retained in the revised manuscript.

• Fig. 3. Precipitation unit? + "Continental outline is represented with the thick black line" or similar.

We will add "**Continental outline is represented with the thick black line.**" to both Figure 2 and Figure 3 for consistency.

• Page 7, lines 2–4. "The annual average … Figs. 2 and 3." Please move these lines and figures into the results section.

David Wade on behalf of the authors

These are not really results per se, they reflect the standard output of the model for the base simulations and so we argue they belong where they are in the method section as a reference for the reader.

• Page 8, line 4. I guess this is "O2 content".
We have changed "O2" to "pO2" to reflect this.

• Page 8, line 11. This sounds unlikely, see for instance Fig. 1 of Royer et al. (doi:10.1130/1052-5173(2004)014<4:CAAPDO>2.0.CO;2).
We stand by the fact that absolute constraints on pre-quaternary $CO_2$ levels are not as good as those in the quaternary. However, we suggest toning down/rewording the sentence to make it clearer that we mean that the upper limits of $CO_2$ are on the order of ~ 100s of Pa, rather than specifically about 100 Pa. The text now reads:
 "however there is growing evidence that $CO_2$ is unlikely to have been significantly higher than the **order of hundreds of Pa** since the radiation of land plants"

• Page 9, line 3. "heuristically". Well, this is obviously "by hand".
Agreed, and as we suggested above this section will be removed.

• Page 9, line 24. What's τs,ebm referring to?
The tau was a typo and should be T. We have corrected this in the modified version.

• Page 10, line 5. Please define "CS" and "CRE".
They are defined on line 3.

• Page 10, lines 14–21. Here and throughout: the text is sometimes difficult to follow because
the authors do not refer to figure panels. Please explicitly include "(Fig. 4b)" etc. when appropriate. Also page 13.
References to Figure 4a-f have been added at relevant points in the first and second paragraphs of the Surface Climate subsection and references to Figure 6a-f in the final paragraph of the Surface Climate subsection.

• Page 10, line 23. "(Figure 4 centre)" > "(Fig. 4 middle column)"
Changed

• Page 10, lines 25–27. "These could be … reduction with height". Is this effect really significant? This could be tested with a flat Earth simulation.
While the result of this sort of idealised study would be of interest, the intention was to note the possible mechanism rather than to quantify the impacts.

• Fig. 4a. Is Panama really open?
The plots show the land sea mask employed by the model. In the case of HadGEM3-AO the ocean is not allowed to mix across the isthmus of Panama but the atmosphere component does not resolve the land bridge hence appears open while the ocean is separated. This is a peculiarity of the differing atmosphere and ocean grids employed by the atmosphere and ocean components.

David Wade on behalf of the authors

• Fig. 4. Please define the "cold month" and the "warm month".
This has been added as "change in the mean gridbox temperature of the coldest month in the monthly mean climatology"

• Table 2.
o Missing data for 4xPI-GEM.
While preparing the manuscript it became evident that the 4x-PI-GEM21 had not been performed so these results are unfortunatley not available.

o The authors may want to include data for their EXP21 experiments in brackets next to their EXP35 results to permit the comparison with Poulsen et al.'s results.
We'd like to thank the reviewer for the suggestion and did implement this, however the resulting table was so crowded as to render it unclear so this has not been updated in the revised manuscript.

• Page 12, line 9. "Comparing the surface temperature (Fig. 4a,b) and precipitation response". Precipitation is showed in the next paragraph.
• Page 12, line 12. "air temperature and precipitation anomalies". Precipitation is showed in the next paragraph.
We'd like to thank the reviewer for noticing these discrepancy. We have moved "Comparing the surface temperature and precipitation response between HadCM3-BL and HadGEM3-AO suggests that the model responses are broadly consistent." and "A gridbox-by-gridbox comparison of annual mean surface air temperature and precipitation anomalies for PI-GEM$^{35}_{10}$ vs PI-CM$^{35}_{10}$ is presented in Fig. S1. The largest discrepancy in surface air temperature response between the two models occurs for the largest temperature changes simulated by HadGEM, which is strongest in Northern Hemisphere polar regions. This could be linked to differences in the representation of polar climate processes between the two models. There is broad consistency in cold and warm-month means (Figure 4a and b) with stronger warming in the cold month mean and terrestrial cooling in the warm month mean." to a new paragraph at the end of the Surface Climate subsection.

• Page 12, line 12. "Fig. S1". Missing supplementary figures? Please check throughout, including on page 15, line 8 + page 18, lines 7–11.
We would like to thank the reviewer for drawing this to our attention. We will ensure that the supplementary figures will be submitted with the revised manuscript - based on the reviewers suggestions we will be including many of these in the main text having removed the Model-data comparison and wind stress sections.

• Page 13, line 1. "representation of polar climate processes between the two models" + amplification by polar ice feedbacks.
The sentence has been adapted to read "This could be linked to differences in the representation of polar climate processes **and amplification by polar ice feedbacks** between the two models"

• Page 13, line 12. Bjerkness compensation.
This has been added to the sentence:
"A northward shift in the ITCZ would be consistent with stronger warming in the Northern Hemisphere **due to Bjerknes compensation (Bjerknes 1964, Broccoli et al. 2006)**."

David Wade on behalf of the authors

• Page 13, line 15. "suggests that pO2 could mediate monsoon climate". Please check in the model output.

Have removed "which suggests that $p\mathrm{O}_2$ could mediate monsoon circulations" as there is no supporting analysis.

• Fig. 6 caption. "Global mean values (mm/day) are offset". Please rephrase. As it is, this suggests that values are really offset, which would be annoying. The text label is offset.

The global mean values are offset from the plot to the top-right. We have rephrased this from "are offset" to "are offset **to the top-right of each plot**" as for figure 4.

• Fig. 7. and Fig. 8:
o Bottom left: What are the dashed lines?
o Bottom right: Grey line is missing in the legend.

The emissivity is indicated by the dashed purple line and the clearsky emissivity is indicated by the solid purple line while the albedo is indicated by the dashed green line and the clearsky albedo is indicated by the solid green line. The dashed green and dashed purple lines are included again in the bottom-left panel so that the reader can contrast the clear-sky vs all-sky components (the remainder being the cloudy-sky component). We infer from the questions that ambiguity arises from the figure caption, hence this is updated to ensure clarity.

The grey line is included in the legend, however is inset in the top-left plot. To ensure that the reader can more easily interpret that the legend entries are common across the plot Figures 7 and 8 have been slightly adjusted:

"Bottom left: Clear-sky emissivity (dark purple) and **clear-sky** albedo (dark green) components of the EBM. **The all-sky components are included for comparison.** Bottom right: Decomposition of EBM into the total clear-sky (blue), cloudy-sky (red) and all-sky (grey) components."

**David Wade on behalf of the authors**

[Figure]

[Figure]

• Fig. 9, caption. "top-of-atmosphere radiative imbalance".

David Wade on behalf of the authors

Thanks - changed "top-of-atmosphere radiative **im**balance"

• Page 18, line 11. "numerically unstable". Any idea why?
An inspection of the model dump files before the crash revealed very high temperatures in the tropics, however a root cause of the instability could not be found. "Runaway" temperatures* have been found in the MPI model at high $CO_2$ (see Heinmann M., PhD Thesis https://pure.mpg.de/rest/items/item_993927_4/component/file_2388525/content Chapter 3 p40). This limit may have been reached in HadCM3-BL in the context of the Maastrichtian climate.

*note this is distinct from a runaway greenhouse effect, the physics of which are typically not properly accounted for in climate models as water vapour is usually assumed to be a minor gas while in a runaway greenhouse will become a major gas and therefore come across similar issues with atmospheric pressure and heat capacity changes. In addition, models would need to better account for water vapour continuum lines than current radiation schemes allow. The rapid runaway temperatures are therefore a model feature rather than an indication that a runaway greenhouse has been reached.

• Section 3.4.
o I suggest changing the title for something more specific like "Response of Permian vegetation to changing O2 levels" since this section really deals with the Permian case study.
At the suggestion of the reviewer we have renamed the section "Response of Permian vegetation to $pO_2$"
o The temperature and precipitation dependence of the dominant PFT simulated in the Permian should also be considered. To what extent are the changes in vegetation cover and type due to changes in temperature and precipitation? Changes in precipitation in the Wu-CM runs (Fig. 6e), in particular, seem to spatially correspond to the expansion of the BLT PFT (Fig. 10). I would like to see a short analysis of the environmental affinities of the main PFTs shown on the maps in Fig. 11. I suspect that temperature and precipitation threshold values may play a more important role than changing O2/CO2 ratios.
Disentangling the myriad of impacts on vegetation cover would be a challenge. The goal of this section is to highlight that $pO_2$ will have implications beyond those of photorespiration, including the impacts of precipitation and temperature changes.

• Fig. 11. The color map is reversed from a to b, which makes it difficult to read. Please revise.
The color maps used are different in all three plots as they represent different variables. If the reader has deuteranopia the scales should still be discernible but will appear to be reversed from a to b.

• Page 19, line 15. "expansive". What does than mean?
Greater in extent - reworded to "broadleaf trees **cover more area** in…"

• Page 20, line 8. I guess it means that the simulated changes in carbon storage on land do not impact the pCO2 level? It would be good to clearly state what the authors mean by "not interactive".

David Wade on behalf of the authors

This is correct. The sentence now reads "the carbon cycle is not interactive **(atmospheric CO₂ is fixed)**" to clarify this point.

 "however, it is likely … equator-to-pole temperature gradient". Please
provide references to support this statement.
This section has been removed from the revised manuscript.

• Section 3.6 "Importance of Wind Stress". I get that atmospheric mass impacts wind stress, which in turn impacts the ocean circulation and the heat transport (see page 4, line 20). Unless I get it wrong, those effects are included in the coupled ocean-atmosphere simulations conducted by the authors, which is a good point. However, I do not understand why the authors test the impact of removing wind stress. In my opinion, this section is off topic and should be deleted and possibly kept for another contribution, which would simultaneously shorten the present manuscript and leave the possibility to conduct a robust analysis of the climate response (including the response of ocean dynamics, the analysis of which is essentially lacking so far). (For this reason, I did not include the minor comments relative to this section in this review).
We have removed the section, as advised by the review in the major comments above.

• Fig. 12, caption. "Proxy data locations (Upchurch et al., 2015) are indicated".
This section has been removed, as advised by the reviewer in the major comments above.

• Fig. 14, caption. What's the unit of precipitation in panel c? Is this an annual mean?
The unit mm day$^{-1}$ has been added to the caption (is already included in the plot). It is annual mean, this has also been added. Now reads:
"**annual mean** precipitation (**mm day$^{-1}$**)"
NB this is now Figure 15 in the revised paper.

 "mainly due to".
We have added the word mainly:
"HadCM3-BL simulates a reduced equilibrium climate sensitivity **mainly** due to changes in longwave cloud feedbacks."

 "The pre-industrial Holocene … of the Archean". Please support this statement with appropriate references.
The references Payne et al (2016), Chemke et al (2016) and Charnay et al (2013) have been added to support the statements made in this sentence.

 So, the implementation of pressure broadening is not the same as in Poulsen et al.? Page 7 line 2 suggests that O2 forcing is analogous to Poulsen et al. Please clearly state what's common between both studies and what's different. Also, I encourage the authors to make their numerical code available for future work (see major comment).
The O₂ forcing *ought* to have been made in an analogous way, however without a deep understanding of the structure of the climate model used and the relevant subcomponents it would not be possible to verify the implementation or identify the commonalities between the implementations. Even if the GENESIS model code and settings were made available, it would be beyond the expertise of the authors to be able to provide a detailed verification. It

is for this reason that we do not believe that releasing model code changes would be useful particularly as the HadCM3-BL and HadGEM3-AO model codebases are under UK crown copyright and the required model changes may vary considerably between models. It is worth noting that while the general approach to changing the $pO_2$ was the same in HadCM3-BL and HadGEM3-AO the specific details are considerably different. Hence we propose that the similarities between the model results and similarities with the more idealised studies elsewhere are what give confidence in the way that the general approach was applied in the specific model versions.

• Page 24, lines 18–21. Since sub-daily model output was not written on disk in Wade et al. model runs and thus not made available for analysis, I suggest deleting this comparison that is not that instructive.

These two sentences have been removed in the revised manuscript.

• Page 24, lines 22–24. Please be cautious: even in a slab model, the continental configuration can impact the ocean heat transport due to the varying ocean area and global climate and thus temperature (which is also impacted by the continental configuration). Another, maybe more robust argument to support the comparison, is that (i) both reconstructions are not so different at first-order and (ii) both simulations provide a relatively close global climate state (compare the mean annual SAT in Poulsen et al. 21% and this study 21% – ca. 18°C vs. 22°C).

We thank for reviewer for this suggestion and have included in this sentence, which we have slightly adjusting to clarify which Cretaceous stage was used.

"Note that the Poulsen et al. 2015 simulations were for an earlier Cretaceous period **(Cenomanian)** than those performed in HadCM3-BL **(Maastrichtian), however the continental configurations and the global mean temperatures are reasonably similar (22.2 °C in HadCM3-BL vs 20.5 °C in Poulsen et al 2015)."**

• Page 24, lines 25–31. The contribution of changing ocean dynamics / deep circulation to the simulated climate changes is addressed for the first time here. I suggest either deleting this unsupported statements or providing clear diagnostics of the changes in ocean dynamics.

We have removed lines 25-31 and lines 32- p25 l1-2 as both paragraphs discuss the offending material.

• Page 25, lines 1–2. The authors may want to refer to Pohl et al. (doi:10.5194/cp-10-2053-2014), who demonstrated the importance of ocean dynamics to simulate Ordovician climate changes.

We'd like to thank the reviewer for this suggestion. We have removed this paragraph as a result of the above comment, so have not included it in the revised manuscript.

• Page 25, lines 5–6. "Increases … high latitudes". Please provide a reference.

Kiehl and Shields 2013 reference has been added.

• Page 25, lines 3–14. The authors may also want to refer to the climatic mechanism demonstrated by Rose and Ferreira (doi: 10.1175/JCLI-D-11-00547.1), which was subsequently invoked by Ladant and Donnadieu (doi:10.1038/ncomms12771) to explain the climate changes observed in their Cretaceous model runs.

David Wade on behalf of the authors

We'd like to thank the reviewer for this insight, while not completely analogous to Rose and Ferreira the enhanced convection at low $pO_2$ is consistent with an atmospheric moistening. Mechanistically, this is more sensitive in a warmer climate due to Clausius-Clapeyron. The subsequent analysis and addition to the manuscript has been included in a response for AR#3.

• Page 25, line 16. "While subsequent experiments have put this in doubt". Please support this statement with a reference.
Wildman et al. 2004 reference added

• Page 25, lines 24–25. "although there is evidence of vegetation which causes C4-like fractionation". During which geological period?
Lower Carboniferous - added to the text as "fractionation **in the Mississippian**, suggesting"

• Page 25, line 31. "Other approaches such as trait based methods". The authors may want to cite Porada et al. (doi: 10.1038/ncomms12113) who applied a trait-based model to simulate the impact of the Ordovician primitive terrestrial vegetation on weathering.
We'd like to thank the reviewer for drawing our attention to this interesting and relevant article and have added to the reference in this sentence: "Other approaches such as trait based methods (Van Bodegom et al., 2012**; Porada et al., 2016**)"

• Page 25, line 32 to page 26, line 2. I suggest deleting this paragraph.
We have removed "We also have not accounted for changes… may be sensitive to atmospheric $pO_2$ (Clarkson et al., 2018)" inclusive.

• Figure 16. The authors previously demonstrated that Poulsen et al.'s implementation of O2 forcing may not be robust. I think this is thus relatively unexpected that they here use those results in their Phanerozoic calculations, even if this may constitute a conservative estimate. I suggest using the results of the current study instead.
The motivation for doing so is that *even if* the climate state was that sensitive to $pO_2$ (which seems unlikely based on previous studies and is also not supported by the proposed study) the differences this causes to global mean temperatures is minor compared to that due to the uncertainty in $CO_2$ content.

• Page 27, lines 17–18. "If pCO2 and pO2 … in the Phanerozoic". Why? Why would cooler climates be associated with higher pO2 levels? I cannot imagine any clear and straightforward explanation to this.
$CO_2$ and $O_2$ are linked by photosynthetic productivity and organic carbon burial - increased carbon burial reduces $CO_2$ and increases $O_2$ ($CO_2 \rightarrow C_{org} + O_2$), see e.g. Montañez 2016 (https://doi.org/10.1073/pnas.1600236113) in the context of $pCO_2$ and $pO_2$ in the Carboniferous-Permian. This mechanism was invoked by Poulsen et al 2015 also.

C. Minor points:
• Page 3, line 2. "(grey shading in Fig. 1)".
Changed
• Page 5, line 13. "which simulates".
Changed

David Wade on behalf of the authors

• Page 6, line 12. Reference formatting: "(Valdes et al., 2017)".
Changed
• Page 6, lines 20–21. Reference formatting.
Changed
• Page 7, line 3. Please use correct experiment names instead of 4 x PI-CM21.
This has been updated and also now includes the sub-letters as the reviewer kindly suggested in an earlier comment:
"**Fig. 1 shows the annual average surface temperatures and Fig. 2 shows the annual average precipitation for the (a) PI-CM$^{21}$, (b) Ma-CM$^{21}$, (c) Wu-CM$^{21}$ and (d) As-CM$^{21}$ simulations**."
• Page 8, line 17. "monotically increasing ozone column". Please rephrase.
The ozone column increases monotonically with $pO_2$ in the Harfoot et al 2007 study, hence was described as such in the manuscript.
• Page 8, line 29. Reference formatting.
Changed
• Page 10, line 19. "Wu-CM" (lower-case).
Changed
• Page 10, line 21. "This suggests that … but is non-linear". Please revise.
We have reworded to "This suggests that the climate response to $pO_2$ variability depends on the background climate state.
• Page 12, line 14. ", which are strongest".
Changed
• Fig. 5, caption. "4XPI-GEM(red)": missing space. See also multiple occurrences on page 18,
lines 1–5.
Changed (fig 5 caption) and updated 5 occurrences page 18.
• Fig. 6 color bar labels. Font size issue leading to overlapping text.
An adjusted figure has been provided in the revised manuscript
• Page 19, line 6. "Fig. 11a".
Changed
• Page 19, line 16. "reduces" > "reduce" (2 occurrences on the same line).
Changed
• Page 20, line 13. Please delete question mark.
Section removed in the revised manuscript
• Page 20, lines 25–26. "These show that across both CO2 contents that increasing". Please rephrase.
Section removed in the revised manuscript
• Page 22, lines 4–5. "has the capacity to alter the radiative budget of the atmosphere and therefore on Earth's climate".
This is verbatim from the original manuscript. Rereading we are suggested to rephrase this text to: "therefore **has implications for** Earth's climate." We hope this aids in the clarity of the sentence.
• Page 22, line 6. "with increasing pO2:".
Colon added
• Fig. 15, caption. Missing space.
Space added
• Page 25, line 3. "also contribute".
This sentence has been removed from the revised manuscript.

• Page 25, line 9. "compared".

This sentence has been removed from the revised manuscript.

• Page 25, line 13. "which may increase lead to". Please revise.

This sentence has been removed from the revised manuscript.

• Page 27, line 7. "PAL". Please write in full or provide meaning.

Changed: PAL → "present atmospheric levels of"

• Page 27, line 16. "When pO2 was higher".

Changed: were → was

**Anonymous Reviewer #3**

The manuscript "Simulating the Climate Response to Atmospheric Oxygen Variability in the Phanerozoic" by Wade et al. presents results from two ocean-atmosphere global circulation models to test the response of temperature, precipitation, and climate sensitivity to variable oxygen levels in earth's past. The primary results are that increasing oxygen levels causes global temperature to increase, precipitation to decrease, and climate sensitivity to change slightly. These results lead the authors to conclude that oxygen is a secondary factor (to CO2, though presumably also to solar luminosity and paleogeography) in earth's climate history. The study is mostly very well done, interesting, and well presented. My comments are mostly minor, and should not impede the eventual publication of the manuscript in Climates of the Past.

The use of two climate models is a strength of this paper, and I commend the authors for the extra effort. However, without a more in-depth discussion of how the models are different and how the differences lead to the responses reported in the paper, the effort falls a little short. It is worth noting and discussing that both models Edwards and Slingo (1996) radiation scheme. What about other physics schemes? Would other non-Hadley models that don't share the same physical parameterizations be expected to have larger differences than these two models?

While the two models are part of the Met Office Unified Model family of models, the two share few physical schemes except for the use of the Edwards and Slingo radiation scheme, although different versions, and the convection scheme is similar (but with a number of updates between HadCM3-BL and HadGEM3-AO). However, there are a number of large differences. It should be noted that the models do not share the same dynamical core and the cloud and precipitation schemes are different. In addition, the ocean model used is different. Whether other models would be expected to have larger differences will depend on the main driver of those differences. If the radiation scheme would cause the greatest difference, this would not be captured here, for instance. Deconvolving the drivers of differences between the model results to individual physics schemes would require a considerable model development effort and the authors are not aware of any effort to perform such a task in the paleoclimate community. While it is possible to rationalise the differences between models (e.g. Lunt et al 2012 doi:10.5194/cp-8-1717-2012), ascribing these changes to particular model components is much more challenging and beyond the scope of our study.

We therefore propose to update the description of model comparison as follows:
", . It is worth noting that HadCM3-BL and HadGEM3-AO are not completely distinct climate models, for instance sharing the Edwards and Slingo 1996 radiation scheme, so this is unlikely to capture the full variability in possible climate model responses. **That the results are**  in reasonable agreement with the 1-D results of Payne et al. 2016**,** who simulated a temperature response between +1.05 and +2.21 $^{\circ}$C depending on assumptions about atmospheric ozone**, gives some confidence in the HadCM3-BL and HadGEM3-AO results.**

One of the most interesting results in the study is the difference in response with geography,

and specifically the fact that the Wuchiapingian simulations show a temperature response that is opposite of the other runs. This is especially interesting in light of the conflicting results from previous models. The authors need to include an analysis and explanation of this result.

The response seen in the Wuchiapingian simulations combined with the lower (less positive) temperature anomaly for 4xPI-GEM$^{35}_{10}$ motivated the transient $CO_2$ doubling experiments which permit an interrogation of the climate sensitivity and the components that contribute towards these. These are provided in the section *Climate Sensitivity* and suggest a higher climate sensitivity at low $pO_2$ which is consistent with the order of temperature anomalies across the experiments (higher in cooler climates, smaller to negative for warmer climates). In addition to this, we have investigated a potential cause of the change and propose that the increase in convection at low $pO_2$ increases atmospheric moistening (see figure overleaf) which has a warming effect analogous to Rose and Ferreira 2013. This would serve to explain not only the changes in climate sensitivity but also the temperature response in the Wuchiapingian.

Changed around p18 l10:
"which tended to cool the low $pO_2$ (**Fig. 11**).
**Unlike the clear sky shortwave effects, the longwave cloud radiative effects seem consistent across the three experiments.**
It should be noted that attempts were made to simulate..."

Added final paragraph of *Climate Sensitivity* subsection:
"**The increase in climate sensitivity appears to be linked to the reduction in temperature anomaly in a warmer climate state. We propose that this is due to more vigorous convection at low $pO_2$ (Goldblatt et al. 2009) leading to an atmospheric moistening (Fig. 12) which causes warming analogously to Rose and Ferreira 2013. This is consistent with the increases in climate sensitivity observed - in a warmer climate the atmosphere can hold more water vapour, so any changes to water vapour will be amplified in their impacts on the radiative budget of the atmosphere. This water vapour feedback is also consistent with the weaker clear sky shortwave radiative effect observed in 2xMa-CM and the temperature response observed in the Wuchiapingian simulations.**"

[Figure]

Caption: "Change in column water vapour in (a) PI-CM$^{35}_{10}$, (b) As-CM$^{35}_{10}$, (c) Ma-CM$^{35}_{10}$ and (d) Wu-CM$^{35}_{10}$. Note the atmospheric drying at high $pO_2$ is enhanced in the warmer climate states of the Wuchiapingian and Maastrichtian and more subdued in the cooler climate states of the Asselian and Holocene."

The manuscript tries to do too much. Section 3.4 is one example (3.5 and 3.6 are others). The discussion of the earth system feedbacks is interesting, but I would have preferred to see it in a standalone study that could do it justice and allow for a fuller discussion of the results and limitations. One shortcoming that the authors do not address is the physiological response of plants to changes in CO2 and O2. How the model handles these changes needs to be described. How well do we know how plants today and in the past responded to changes in atmospheric composition? Recent literature also indicates that changes in soil respiration may be as important as changes in plant respiration. How is this handled in the model?

The use of modern plant functional types for past climates is a key limitation of this section (as openly stated on p19 l2-4 of the OM), however is a necessary evil given the nature of this type of climate modelling. State-of-the-art offline methods (not coupled to climate models) use trait-based approaches (e.g. Porada et al. doi: 10.1038/ncomms12113, recommended by AR#2) that model plant physiological strategies. This is becoming the recommended way to simulate vegetation over the traditional PFT framework ( see e.g. https://doi.org/10.1177/0309133315582018), however this will take time to filter into coupled climate models. Soil respiration is not treated in the model, however it is worth noting that due to the carbon cycle not being interactive it would not affect the results of the study (treatment of soil carbon in MOSES, now JULES is relatively recent, see doi:10.5194/gmd-10-959-2017). In MOSES2.1, atmospheric oxygen affects the photorespiration compensation point so the text has been updated to specify this:
"accounting for a number of factors including atmospheric oxygen content, **which affects the photorespiration compensation point (Clark et al. 2011)."**

Section 3.5 and the discussion of other mechanisms for producing warm climates is really a distraction from the main focus of the paper. The model-data comparison is not particularly rigorous and not necessary, and the discussion of warming mechanisms is incomplete and doesn't reference many important studies. Both sections should be

David Wade on behalf of the authors

deleted.

We thank the reviewer for this suggestion and have removed the model-data section from the revised manuscript. We have also removed the paragraph beginning "Increased oxygen content may also contribute to explaining…"

Section 3.6 on the influence of wind stress is interesting, but not very insightful without a proper analysis of the explanation for the differences between runs. This section should be removed or (preferably) expanded. How does the total heat transport differ between these runs with and without wind stress?

We thank the reviewer for this suggestion and have removed this section from the revised manuscript.

One of the main results of the paper is that the response to changes in O2 is very much a function of cloud feedbacks (e.g. Section 3.2). How robust then are the results? How do cloud feedbacks in HadCM and Had GEM3 compare to each other and to other models? This major point is not discussed in the Discussion or presented in the Conclusions.

The cloud feedbacks due to $pO_2$ changes have not been assessed in other climate models. Studies of cloud feedbacks in the context of climate models mostly relate to $CO_2$ forcing. The closest analogue to the radiative changes associated with $pO_2$ variability would be solar geoengineering due to the offset of shortwave and longwave radiation. A slightly earlier version of the HadGEM3 model (HadGEM3-ES v6.6.3 vs HadGEM3-AO v7.3) has longwave cloud feedbacks broadly in line with other climate models (Russotto et al. 2018, see https://doi.org/10.5194/acp-18-11905-2018)

P. 3, L. 17. "which is consistent with the long-term sensitivity of the Earth system to CO2 changes. . ." I don't understand this comment. The fact that the CO2 range is constrained should not have an influence on the climate system sensitivity to CO2.

We have removed this last part of the sentence starting on line 16 page 3.

P. 16, L. 6-7. Please state the climate sensitivity of HadGEM3-AO and HadCM3-BL.

This information and references to the values have been added to the text as "For reference, HadGEM3 has a climate sensitivity of +3.6 $^{\circ}$C (Nowack et al. 2015) and HadCM3 has a climate sensitivity of +3.1 $^{\circ}$C (Johns et al 2006)."

P. 18, L. 1. "The clear-sky longwave radiative flux changes are higher in PI2X-CM. . ." That's not what I see in Fig. 9a. Is there a typo here, or am I misinterpreting something?

The change is quite small, so this has been reworded to "radiative flux changes are **slightly** higher".

P. 18, L. 8. "For Ma-CM, this value is much larger." This is an interesting result that is not intuitive. The authors should provide a fuller explanation of the large change in sensitivity with this paleogeography and include the figure in the main text.

These figures have moved to the main text. The cause of this difference is addressed in the comment above.

David Wade on behalf of the authors

[revised manuscript text omitted]
\text{O}_2$ is increased in As-CM ($-338\,\text{Pg C}$) and higher as $p\text{O}_2$ is increased in Wu-CM ($+379\,\text{Pg C}$). This is dominated by changes in the tropics (in agreement with Beerling and Berner 2000), where broadleaf trees  cover more area in Wu-CM[35]. Cooler terrestrial tropical temperatures, particularly in the warm month (Figure 4e)  reduce the $p\text{O}_2$ inhibition of Ru-

[Figure]

**Figure 14.** As-CM$_{10}^{35}$ (left) and Wu-CM$_{10}^{35}$ (right) anomalies for (a) net primary productivity, (b) total carbon storage and (c) water use efficiency.

bisco and  reduce the rate of respiration by vegetation and soils (Long, 1991; Beerling and Berner, 2000).
5

As these simulations are fully coupled and changes to oxygen content affect temperatures, radiation and precipitation it is challenging to explore all the possible contributions to differences between these results and the more idealised Beerling and Berner (2000) simulations. However, there is general agreement that changes occur in the signs of the response of NPP and total carbon storage. This supports the conclusions of Beerling and Berner (2000) that high $p$O$_2$ in the early Permian may have
10   played an important role in the evolution of plants. Note that while the atmosphere and vegetation are coupled in the physical sense, the carbon cycle is not interactive

15

**3.5**

While the changes to global-mean surface temperature (GMST) from changes in (atmospheric $CO_2$ is fixed) so determining the impacts of atmospheric $p$ are less substantial than for large changes in , regional changes are comparable to other smaller changes which have been widely investigated such as changes to topography or the differences between forcing and cloud albedo modification (Carlson and Caballero, 2017). This raises the question whether oxygen content could reasonably alter the agreement between models and proxy data? For this we employ the Maastrichtian experiments as there is a considerable quantity of widely used proxy data for the Maastrichtian (Upchurch et al., 2015). In addition to the Ma-CM[35], Ma-CM[21] and Ma-CM[10] experiments, further experiments (the Ma2x-CM[35]* and Ma2x-CM[10]*) were iterated for 1000 model years and the final 50 years were analysed. O2 on the carbon cycle remains an outstanding problem.

Figure ??a shows the annual mean surface air temperature (SAT) simulated for the Ma-CM[21] case along with the locations of the proxy data employed for comparison with the model in Fig. ??b. As is commonly observed amongst many climate models, HadCM3-BL struggles to simulate the high latitude warmth indicated by proxy reconstructions. A consideration of the seasonality of the proxies could reconcile some of these differences (Figure ??b, grey vertical bars), however it is likely that a number of factors such as model deficiencies or climate feedbacks such as convective clouds or cloud droplet radius changes may play a role in the shallower equator-to-pole temperature gradient. The normalised mean bias, root mean square error, normalised mean bias factor and normalised mean absolute error factor (Yu et al., 2006) of the Ma-CM[35], Ma-CM[21], Ma-CM[10], Ma2x-CM[10]* and Ma2x-CM[35]* experiments are shown in Table ??. These show that across both contents that increasing the oxygen content leads to a reduction in all bias metrics against the Upchurch et al. (2015) data, however doubling the content led to the largest improvement in bias scores.

Comparison between the simulated annual mean surface air temperature and the Upchurch et al. (2015) reconstructed surface air temperature. NMB: normalised mean bias. RMSE: root mean square error. NMBF: normalised mean bias factor. NMAEF: normalised mean absolute error factor (Yu et al., 2006). Best performing simulation for a particular metric is indicated in bold. Experiment NMB RMSE / NMBF NMAEF Ma-CM[35] −0.247.92−0.310.41 Ma-CM[21] −0.288.68−0.390.48 Ma-CM[10] −0.288.83−0.400.50 Ma2x-CM[35]* **−0.045.82−0.040.25** Ma2x-CM[10]* +0.076.22+0.070.26

Surface air temperature anomalies for (a) $\tau$PI-CM*−PI-CM[21], (b) $\tau$Ma-CM*−Ma-CM[21], (c) $\tau$Wu-CM*−Wu-CM[21], (d) $\tau$As-CM*−As-CM[21]

**3.5 Importance of Wind Stress**

Saenko (2009) assessed the contribution of wind stress to global climate in the Canadian Centre for Climate Modeling and Analysis model by setting the wind stress experienced by the ocean to zero and simulated a reduction in global mean surface temperature by 8.7. Similar simulations were performed with HadCM3-BL ($\tau$PI-CM*, $\tau$Ma-CM*, $\tau$Wu-CM* and $\tau$As-CM*) and iterated for 500 model years. This is insufficient to achieve an equilibrium climate response, however as the aim is to investigate the relative magnitude of the changes this was considered adequate to explore this sensitivity study.

The impacts of removing wind stress are shown in Fig. ??. HadCM3-BL simulated a −7.3surface air temperature change, slightly weaker than Saenko (2009) however it should be noted that HadCM3-BL was iterated for substantially longer (500 years vs 100 years). $\tau$As-CM* was the most sensitive to the removal of wind stress, with a −15.7SAT change. By contrast,

[Figure]

**Figure 15.** 21 %–10 % $O_2$ anomalies for Poulsen et al. (2015) simulations. (a) Annual mean, cold month mean and warm month mean surface air temperature difference. (b) Change to diurnal cycle and (c) annual mean precipitation (mm day$^{-1}$).

~~the warmest climates showed more muted cooling in response to the removal of wind stress forcing. $\tau$Ma-CM\* shows an SAT anomaly of –2.7while $\tau$Wu-CM\* shows an SAT anomaly of –3.51. This suggests that the climate response to wind stress changes is likely to depend on the ocean configuration and the background climate – warmer climates of the Wuchiapingian and Maastrichtian appear to be much less sensitive to wind stress forcing. This could be due to the lower meridional temperature~~
5

**4   Discussion**

Through its impact on atmospheric mass, oxygen content has the capacity to alter the radiative budget of the atmosphere and therefore  has implications for Earth's climate. These simulations suggest that the interactions between radiative and dynamical feedbacks lead to some consistent climatic changes in HadCM3-BL with increasing $pO_2$:

10  – Reduction in the seasonal cycle in surface air temperature.

– Reduction in equator-to-pole temperature gradient.

– Reduction in global precipitation.

HadCM3-BL simulates a reduced equilibrium climate sensitivity mainly due to changes in longwave cloud feedbacks. HadGEM3-AO results also support a reduced sensitivity to $CO_2$ content at high $pO_2$. The pre-industrial Holocene results are supported
15  by 1D radiative convective simulations (Payne et al., 2016), 2D model simulations (Chemke et al., 2016) and slab ocean 3D

[Figure]

**Figure 16.** 1D-energy balance decomposition analogous to Fig. 7 for the 21–10 % $O_2$ Poulsen et al. (2015) simulations.

model simulations of the Archean (Charnay et al., 2013). This raises a discrepancy with the Poulsen et al. (2015) study, which simulated a reduction in global mean surface temperature when increasing oxygen content in the GENESIS model. Figure 15a shows the surface air temperature change between the 10 % and 21 % Cenomanian (100.5–93.9 Ma) simulations from the Poulsen et al. (2015) study. These show a –2.04°C change in the annual mean. To understand the mechanisms behind this,

5   we performed the 1D-energy balance decomposition on the Poulsen et al. (2015) Cenomanian 21–10 % model output. The results are shown in Fig. 16. This shows that the cloudy-sky contribution to the temperature change dominates the climate response, contributing –1.45°C. However, the clear sky contribution is also negative (–0.60°C) including both clear-sky emissivity (–0.58°C) and clear-sky albedo (–0.11°C). This appears to support the argument that tropical cloud feedbacks explain the discrepancy between the Poulsen et al. (2015) simulations and results of 1D radiative convective models (Goldblatt, 2016),

10  however this cannot be the only factor. An increase in pressure broadening of absorption lines would be expected to lead to a positive contribution from the clear-sky emissivity. This suggests that cloud feedbacks alone cannot explain the discrepancy and that the implementation of pressure broadening may play a role in the anomalous Poulsen et al. (2015) response. In addition, changes to the seasonal cycle (Figure 15a) simulated by Poulsen et al. (2015) are also inconsistent with the HadGEM-AO and HadCM3-BL results, in which all simulations led to a reduced seasonal cycle as $pO_2$ increases. The Poulsen et al. (2015)

15  Cenomanian simulations actually simulates a larger seasonal cycle at high $pO_2$ which is challenging to reconcile with the radiative and physical processes.

 Note that the Poulsen et al. (2015) simulations were for an earlier Cretaceous period (Cenomanian) than those performed in HadCM3-BL

5  (Maastrichtian), however the continental configurations and the global mean temperatures are reasonably similar (22.2 °C in HadCM3-BL vs 20.5 °C in Poulsen et al. (2015)).

The simulations presented in here suggest that perturbations to the wind-driven ocean circulation by increasing atmospheric mass leads to warmer temperatures, particularly at high latitudes. The magnitude of the results varies depending on the precise

10  continental configuration and background climate state. Gyre circulations vary between the preindustrial and the Maastrichtian and Asselian case studies. Given the importance of the wind-driven ocean circulation response this suggests that a 3D representation of ocean circulation is necessary in order to capture the temperature response to atmospheric mass changes. It should be noted however that Charnay et al. (2013) simulated higher surface temperatures for the early Earth at high atmospheric mass with a slab ocean model.

15  The use of 3D oceans is now widespread in the palaeoclimate community, however this is not widely used in the exoplanet/early earth community (e.g. Kilic et al. 2017) and for early Earth studies such as the Archean (e.g. Charnay et al. 2013). While boundary conditions for these studies are sparse or in some cases non-existent the additional uncertainty associated with using a slab ocean should be considered. AO-GCM studies remain the best way to assess the complex coupling between potentially competing radiative and dynamical effects.

20  ~~Increased oxygen content may also contribute to explaining the very low temperature gradients for hothouse climates in the Phanerozoic – the "shallow gradients paradox" (Huber and Caballero, 2011). However, there are other mechanisms which could lead to similar changes. Increases in the effective radii of liquid clouds leads to considerable warming, particularly at high latitudes. While the tropics also warm the equator-to-pole temperature gradient is reduced. Abbot and Tziperman (2008) describe a convective cloud feedback which warms the high latitudes, particularly in winter. Upchurch et al. (2015) found that~~

25  ~~CCSM3 is able to reasonably simulate the shallow Maastrichtian temperature gradient when the effective radii of cloud droplets was set to 17 globally (comapred to typical values today around 8 over land surfaces and 14 over ocean surfaces and 17 only observed for the most pristine of clouds in the current atmosphere), however this is hard to reconcile with the high primary productivity likely in hothouse climates and the large contributions of biogenic sources of volatile organic compounds and ammonia in the pre-industrial atmosphere (Gordon et al., 2017). In addition, the seasonality of cloud droplet changes is likely~~

[revised manuscript text omitted]

Upchurch, Jr, G. R., Kiehl, J., Shields, C., Scherer, J., and Scotese, C.: Latitudinal temperature gradients and high-latitude temperatures during the latest Cretaceous: Congruence of geologic data and climate models, Geology, 43, 683, https://doi.org/10.1130/G36802.1, 2015.

Valdes, P. J., Armstrong, E., Badger, M. P. S., Bradshaw, C. D., Bragg, F., Crucifix, M., Davies-Barnard, T., Day, J. J., Farnsworth, A., Gordon, C., Hopcroft, P. O., Kennedy, A. T., Lord, N. S., Lunt, D. J., Marzocchi, A., Parry, L. M., Pope, V., Roberts, W. H. G., Stone, E. J., Tourte, G. J. L., and Williams, J. H. T.: The BRIDGE HadCM3 family of climate models: HadCM3@Bristol v1.0, Geoscientific Model Development, 10, 3715–3743, https://doi.org/10.5194/gmd-10-3715-2017, 2017.

Van Bodegom, P. M., Douma, J. C., Witte, J. P. M., Ordoñez, J. C., Bartholomeus, R. P., and Aerts, R.: Going beyond limitations of plant functional types when predicting global ecosystem-atmosphere fluxes: exploring the merits of traits-based approaches, Global Ecology and Biogeography, 21, 625–636, https://doi.org/10.1111/j.1466-8238.2011.00717.x, 2012.

Watson, A., Lovelock, J. E., and Margulis, L.: Methanogenesis, fires and the regulation of atmospheric oxygen, Biosystems, 10, 293–298, https://doi.org/10.1016/0303-2647(78)90012-6, 1978.

White, A. A. and Bromley, R. A.: Dynamically consistent, quasi-hydrostatic equations for global models with a complete representation of the Coriolis force, Quarterly Journal of the Royal Meteorological Society, 121, 399–418, https://doi.org/10.1002/qj.49712152208, 1995.

Wildman, R. A., Hickey, L. J., Dickinson, M. B., Berner, R. A., Robinson, J. M., Dietrich, M., Essenhigh, R. H., and Wildman, C. B.: Burning of forest materials under late Paleozoic high atmospheric oxygen levels, Geology, 32, 457, https://doi.org/10.1130/G20255.1, 2004.

Williams, K., Copsey, D., Blockley, E., Bodas-Salcedo, A., Calvert, D., Comer, R., Davis, P., Graham, T., Hewitt, H., Hill, R., et al.: The Met Office global coupled model 3.0 and 3.1 (GC3. 0 and GC3. 1) configurations, Journal of Advances in Modeling Earth Systems, 10, 357–380, 2018.

Wilson, J. P., Montañez, I. P., White, J. D., DiMichele, W. A., McElwain, J. C., Poulsen, C. J., and Hren, M. T.: Dynamic Carboniferous tropical forests: new views of plant function and potential for physiological forcing of climate, New Phytologist, 215, 1333–1353, https://doi.org/10.1111/nph.14700, 2017.

Yu, S., Eder, B., Dennis, R., Chu, S.-H., and Schwartz, S. E.: New unbiased symmetric metrics for evaluation of air quality models, Atmospheric Science Letters, 7, 26–34, https://doi.org/10.1002/asl.125, 2006.